# Evaluating the Impact of Enhanced Horizontal Resolution over the Antarctic Domain Using a Variable-Resolution Earth System Model

Rajashree Tri Datta[1], Adam Herrington[2], Jan T.M. Lenaerts[1], David P. Schneider[2,3], Luke Trusel[4], Ziqi Yin[1], Devon D. Dunmire[1]

[1]Atnospheric and Oceanic Sciences Dept., University of Colorado, Boulder, Boulder, CO 80309, USA
[2]Climate and Global Dynamics Laboratory, National Center for Atmospheric Research, Boulder, CO 80305, USA
[3]Cooperative Institute for Research in Environmental Sciences, University of Colorado – Boulder, Boulder, CO 80309 USA
[4]Department of Geography, Pennsylvania State University, University Park, PA 16802, USA

*Correspondence to*: Rajashree Tri Datta (Tri.Datta@gmail.com)

**Abstract.** Earth System Models are essential tools for understanding the impacts of a warming world, particularly on the contribution of polar ice sheets to sea level change. However, current models lack full coupling of the ice sheets to the ocean, and are typically run at a coarse resolution (1 degree grid spacing or coarser). Coarse spatial resolution is particularly a

problem over Antarctica, where sub-gridscale orography is well-known to influence precipitation fields, and glacier models require high-resolution atmospheric inputs. This resolution limitation has been partially addressed by regional climate models (RCMs), which must be forced at their lateral and ocean surface boundaries by (usually coarser) global atmospheric datasets, However, RCMs fail to capture the two-way coupling between the regional domain and the global climate system. Conversely, running high spatial resolution models globally is computationally expensive, and can produce vast amounts of

data.

Alternatively, variable-resolution grids can retain the benefits of high resolution over a specified domain without the computational costs of running at a high resolution globally. Here we evaluate a historical simulation of the Community Earth System Model, version 2, (CESM2) implementing the spectral element (SE) numerical dynamical core (VR-CESM2)

with an enhanced-horizontal-resolution (0.25°) grid over the Antarctic Ice Sheet and the surrounding Southern Ocean; the rest of the global domain is on the standard 1° grid. We compare it to 1° model runs of CESM2 using both the sSE dynamical core and the standard finite-volume (FV) dynamical core, both with identical physics and forcing, including prescribed SSTs and sea ice concentrations from observations. Our evaluation reveals both improvements and degradations in VR-CESM2 performance relative to the 1° CESM2. Surface mass balance estimates are slightly higher, but within one

standard deviation of the ensemble mean, except for over the Antarctic Peninsula, which is impacted by better-resolved surface topography. Temperature and wind estimates are improved over both the near-surface and aloft, although the overall correction of a cold bias (within the 1° CESM2 runs) has resulted in temperatures which are too high over the interior of the ice sheet. The major degradations include the enhancement of surface melt as well as excessive cloud liquid water over the

ocean, with resultant impacts on the surface radiation budget. Despite these changes, VR-CESM2 is a valuable tool for the analysis of precipitation and surface mass balance, and thus constraining estimates of sea level rise associated with the Antarctic Ice Sheet.

## 1 Introduction

The Antarctic Ice Sheet (AIS) contains the largest amount of freshwater on Earth and, with the Greenland Ice sheet, is the greatest potential source of future sea level rise (Pan et al., 2021). Satellite observations have captured net mass loss of the AIS (Shepherd et al., 2019) which has, since 1993, accounted for approximately 10% of observed global mean sea level rise (Oppenheimer et al., 2019). Mass loss follows from a negative difference between the surface mass balance (SMB), consisting of gain by snowfall and loss through sublimation and runoff, and the discharge of solid ice (D) across the grounding line (SMB-D < 0). While SMB over the floating ice shelves that ring the continent does not directly affect sea level rise, surface melt can impact the stability of ice shelves, leading to rapid drainage and hydrofracture (Robel and Banwell, 2019; Banwell et al., 2013) or erosion of the firn layer, which reduces ice shelf resilience to atmospheric warming (Gilbert and Kittel, 2021). The loss of the buttressing effect of ice shelves can induce a speedup in tributary glaciers on the grounded ice sheet, which contributes to sea level rise (Hulbe et al., 2008).

In the current era (since 1979), mass loss over Antarctica has largely resulted from ocean-driven processes, whereby warm ocean waters erode ice shelves from below (Rignot et al., 2019), while mass loss due to meltwater runoff has been negligible (Lenaerts et al., 2019). However, future warming scenarios will be strongly affected by the interplay between mass loss at the margins, potentially enhanced surface melt, e.g. the "Greenlandification of Antarctica" (Trusel et al., 2018) and greater snowfall associated with higher temperatures (Kittel et al., 2021). Other components of the earth system can also affect SMB, including surrounding sea ice, which can impact the intrusion of precipitation (Wang et al., 2020) as well as provide a buttressing effect for tributary glaciers (Wille et al., 2022).

Earth Systems Models (ESM) are coupled climate models which capture the interaction between many systems, e.g. ice sheets, atmosphere, oceans and land. ESMs such as the Community Earth System Model 2 (CESM2) are typically run at coarse horizontal resolutions (e.g. 1°), thus requiring higher-resolution regional climate models (RCMs), forced by coarser reanalysis or ESM outputs at the boundaries, to resolve relatively small-scale physical processes such as orographic precipitation. Variable-resolution (VR) grids combine the advantages of high resolution over focus areas, with coarser resolution elsewhere in the global domain, and therefore avoids the computational costs of running a high-resolution model globally. In contrast to RCMs, a variable-resolution grid can maintain the two-way interaction between the high-resolution domain and the larger global climate system, maintaining an integrated model framework (Huang et al., 2016). Previous work has achieved higher resolution over Antarctica by stretching the global grid; this differs from the refined-resolution method presented here in that, for a stretched grid, enhanced resolution in one domain is achieved by reducing resolution over the remainder of the globe (Krinner et al, 1997a, 1997b, 2007, 2014).

VR-CESM2 employs a spectral element dynamical core (SE; Dennis et al., 2012; Lauritzen et al., 2018) and the version used in this study uses CAM6 physical parameterizations (referred to as physics hereafter). Previous work has demonstrated that VR-CESM captures both the climate of the lower-resolution finite volume (FV) dynamical core while also capturing the high-frequency statistics of higher-resolution models, although biases remain (Gettelman et al., 2019). VR-CESM has also compared favorably to high-resolution regional climate model simulations over California (Rhoades et al 2016; Huang et al., 2016) and the Peruvian Andes (Bambach et al 2021). Finally, surface mass balance and related variables have been evaluated over Greenland, with increasingly higher-resolution VR-CESM runs over an abbreviated period, finding that enhanced resolution produced a positive SMB bias, generally constituting an improvement (van Kampenhout et al., 2019). We note that all of the simulations mentioned thus far have used CAM5 physics, whereas this work over Antarctica as well as a comparison between multiple variable-resolution runs over Greenland (Herrington et al., 2022) are using the newer CAM6 physics.

Within this study, we evaluate a variable-resolution grid over the Antarctic domain, which we refer to as ANTSI (ANTarctic Sea Ice), as it includes both the Antarctic Ice Sheet and surrounding Southern Ocean domain, with a higher spatial resolution covering the maximum mean extent of sea ice in the historical (1979-2014) era. In addition to the effects of enhanced resolution over the ice sheet, the enhanced spatial resolution over the ocean can alter temperature and moisture fluxes, and strongly impact surface mass balance through the incursion of storms. Thus, the evaluation will focus partially on surface mass balance and energy balance components on the surface, but also the effect of enhanced resolution on the larger Southern Ocean domain. For an evaluation of how the improvements to CAM6 within CESM2 impact surface mass balance and the near-surface climate with the 1° FV dynamical core, we refer the reader to Dunmire et al., 2022. As compared to CESM1 with CAM5 (at the same resolution), CESM2/CAM6 showed improvements in near-surface temperature, windspeed, and surface melt, but excessive precipitation (Dunmire et al., 2022).

Here we will focus specifically on the combined impact of enhanced resolution and SE dynamical core over the Antarctic domain. We will compare these results to similar model runs with identical physics and forcing, but with a lower resolution and using the FV dynamical core. Model outputs will be compared between these model runs as well as to reanalysis, in-situ weather station data and regional climate models.

Data and Methods are described in Section 2, Results in Section 3 (including surface mass balance in Section 3.1, followed by drivers of enhanced moisture in Section 3.2 and finally impacts on the radiation balance in Section3.4), and Discussion and Conclusions in Section 4.

## 2 Data and Methods

### 2.1 Model: CESM2

The ESM used here, the Community Earth System Model, version 2.2 (CESM2.2; Herrington et al., 2022; Danabasoglu et al., 2020) couples several climate components, including the atmosphere (Community Atmosphere Model

(CAM6; Neale et al., 2010), ocean (by the Parallel Ocean Program (POP2; Smith et al., 2010)) and sea ice (the Community Ice CodE for sea ice (CICE; Hunke & Lipscomb, 2010)), land (The Community Land Model (CLM; Lawrence et al., 2011)), and ice sheets (Community Ice Sheet Model (CISM; Lipscomb et al., 2013)). Currently, CISM is only active for the Greenland Ice Sheet, with the coupling for the AIS in development.

CAM6 physics in CESM2 is described in detail in Gettelman et al 2019. The SE VR configuration is described in Herrington et al. (2022).

### 2.1.1  The ANTSI grid

The ANTSI (ANTarctic Sea Ice) grid (Fig. 1a) is implemented in CESM2.2, with 32 atmospheric layers. The interior domain has a resolution of 0.25°, while a nominal 1° (standard) grid in the exterior, with intermediate-resolution points along the boundaries, which extend as far north as 55°. These simulations use the AMIP setup (Atmospheric Model Intercomparison Project; Gates et al., 2019), which is a protocol where land and atmosphere components are freely evolving, but use prescribed sea-surface temperatures (ERSSTv5 – Huang et al., 2017) and sea-ice (HadISST- Rayner et al., 2003), in this case for the period 1979 to 2014.

In addition to the standard monthly outputs, we produce daily and 3-hourly outputs for selected variables for the surface and 3-hourly outputs for selected variables for both the atmosphere and ice sheet surface. These higher-temporal resolution outputs will be used for future analysis of atmospheric rivers and forcing of firn models. The computational cost is approximately 23,000 core hours per model year (on the NCAR-Wyoming Cheyenne Supercomputer), and the enhanced storage requirements will scale linearly with the number of cells. For example, the standard 1° FV grid has 55296 cells, whereas the ANTSI grid contains 117398 gridcells (thus requiring ~double storage of a 1° model)

Elevation over Antarctica (Fig. 1b) is highest in the interior, and highly variable at the margins, and ANTSI captures finer resolution than the 1° FV grid over both the ocean and ice sheet. Thus, the enhanced resolution of ANTSI better captures topography along the coasts, with some regions of the grounded ice sheet resolving over 400m higher at the ice sheet margin (e.g. the spine of the Antarctic Peninsula) while ice shelves show slightly lower elevations (Fig. 1b,d).

[Figure 1]

### 2.1.2 CESM2 AMIP (GOGA) – Used for evaluation

This evaluation primarily focuses on a comparison of ANTSI with the Global Atmosphere-Global Ocean (GOGA) ensemble, a 10-member set of 1° prescribed-SST experiments with CAM6 conducted by the CESM Climate Variability and Change Working Group (https://www2.cesm.ucar.edu/working_groups/CVC/simulations/cam6-prescribed_sst.html) that generally follows what is commonly referred to as the "AMIP" experimental design. In GOGA-AMIP (hereafter using "AMIP" for short), time-varying SSTs and sea ice concentration are taken from observations; namely, ERSSTv5 (Huang et al., 2017) and HadISST (Rayner et al., 2003), respectively. External radiative forcings use the CMIP6 historical protocol

(Eyring et al., 2016) similarly to our ANTSI experiment. For most of our evaluation, we use the ensemble mean of AMIP, except as noted below.

One difference between AMIP and ANTSI that could be important for our evaluation is the dynamical cores of CAM6; ANTSI uses SE while AMIP uses FV. While it is beyond the scope of this work (and our resources) to conduct a full AMIP-style ensemble with CAM6 and the SE dynamical core, we did conduct two, one-member runs with the FV and SE dynamical cores for the 1979-1998 period in the course of model development of the VR-CESM2 (Herrington et al., 2022). These runs are compared and discussed in section 3.3.3 in order to help determine whether differences between ANTSI and GOGA are due to dynamical core or resolution.

### 2.1.3 Calculation of Surface Mass Balance

Surface mass balance (SMB, in gigatons, or GT yr$^{-1}$) is calculated for both AMIP and ANTSI as follows (adapted from Lenaerts et al., 2019):

SMB = Precipitation - Runoff  - Sublimation/Evaporation          (1)

Precipitation =  Snow + Rainfall          (2)

Runoff = Melt + Rainfall + Condensation - Refreezing - Retention          (3)

We note that the ""Refreezing" and "Retention" terms apply to both "Melt" and "Rainfall". For integrated surface mass balance elements as well as snowmelt comparisons, we remap from the finer resolution to the coarser-resolution grid using the Earth Systems Modeling Framework (ESMF), described at http://earthsystemmodeling.org/, using their first-order conservative  method and accounting for the percentage of the gridcell that is covered by glacier.

For all other comparisons, in order to preserve the spatial characteristics of the finer resolution, we regrid from a lower resolution to the finer ANTSI resolution using the ESMF bilinear method. The ice sheet mask is provided by Zwally et al. (2002).

### 2.2 Datasets Used for Evaluation

We compare surface mass balance values to those from RACMO2.3p2 (van Wessem et al., 2018), RACMO2.3p2 hereafter. RACMO2.3p2 is a hydrostatic regional climate model (RCM) forced at the boundaries with ERA-Interim, with a

multi-layer snow scheme (Ettema et al., 2010), a drifting snow scheme across the ice sheet (Lenaerts et al., 2010, 2012), a sophisticated albedo scheme (Kuipers Munneke et al., 2011) and orography derived from Bamber et al. (2009).

Monthly-averaged ERA5 (Hersbach et al., 2020) data is used here for the 1979-2014 period for near-surface temperature, wind and energy balance components as well as temperature, geopotential height, moisture and wind profiles over the
Southern Ocean. This dataset assimilates observations into a numerical weather prediction model, producing data at an hourly resolution from 1950 to the present, a horizontal resolution of 31km, 139 vertical levels (to 1 Pa) and an assimilation of several data products for an elevation (ECMWF, 2016).

For temperature, we note that ERA5 has been shown to exhibits a slight cold bias in winter in comparison to Automatic weather station (AWS) data (excepting the Antarctic Peninsula) and a warm bias in summer (excepting West Antarctica)
(Zhu et al., 2021).

AWS observations are used here for near-surface temperature and wind speed, retrieved from a collection of 133 automatic AWSs previously evaluated over the AIS (Gossart et al., 2019). These were limited to stations where greater than 10 years of near-surface temperatures and windspeeds were available and where elevation differences between the station and the
resolved elevation (from either AMIP or ANTSI) were smaller than 100 m. Additionally, we apply a temperature adjustment to the AMIP or ANTSI elevation for each AWS using a temperature lapse rate of 6.8°C/km (Martin and Peel, 1978). This final set is divided into bins for elevation, with 21 stations > 500m, one station between 500 and 1000 m above sea level (a.s.l.), 3 stations between 1000 and 2000 m, 8 stations between 2000 and 3000 m and 4 stations above 3000 m.

For comparisons to melt observations, we use melt fluxes calculated empirically from radar backscatter from the QuikSCAT (QSCAT) satellite (Trusel et al., 2013), available at 27.2 km2.  For comparisons to surface mass balance, we use the AIS SMB reconstruction from Medley and Thomas (2019), which interpolates ice core SMB records based on covariance patterns in atmospheric reanalysis (here, using the MERRA2 version of their reconstruction).

**3 Results**

**3.1 Surface Mass Balance**

**3.1.1 Integrated Values for Surface Mass Balance**

SMB is lowest in the high elevations of the AIS desert interior, where storms rarely penetrate, and much higher at the coastal margins (Mottram et al., 2021). ANTSI annual mean (1979-2014) AIS integrated SMB is 2668 GT yr$^{-1}$, with a temporal
standard deviation of 123 GT yr$^{-1}$, as compared to the AMIP SMB mean over the same period of 2515 GT yr$^{-1}$, with an

ensemble standard deviation of 53 GT $yr^{-1}$. This places integrated ANTSI SMB values within one standard deviation of the AMIP SMB mean. This is higher than the 2483 GT $yr^{-1}$ ensemble mean calculated for 1979-2018 (the highest calculated being HIRHAM5 0.44 at 2752 GT $yr^{-1}$) by Mottram at al., (2021). Regions here (Fig. 1d) are divided by land cover (grounded, ice shelf, combined) and are calculated using basins described in Zwally et al. (2002), with all SMB values and trends listed in Table 1. Mean integrated ANTSI SMB values are slightly higher, but within one standard deviation of, the AMIP ensemble mean, for East Antarctica and West Antarctica. ANTSI SMB values for the Antarctic Peninsula are greater than one standard deviation for the AMIP mean (151 ± 13 GT $yr^{-1}$ for ANTSI vs 113±6 GT $yr^{-1}$ for AMIP), driven by enhanced SMB over the grounded ice sheet (106 ± 10 GT $yr^{-1}$ for ANTSI vs 72±4 GT $yr^{-1}$ for AMIP).

[Table 1]

Regionally integrated SMB from ANTSI over the 1979-2014 period is provided in Figure 2. The figure shows several statistically significant ($p < 0.05$) trends, including a -0.80 GT $yr^{-2}$ negative trend in SMB over East Antarctic ice shelves which is not replicated by AMIP (Fig. 3b), as well as a positive trend over the grounded ice sheet of the AP for both AMIP (0.24 GT $yr^{-2}$) and ANTSI (0.51 GT $yr^{-2}$) .

### 3.1.2 Spatial Patterns for Surface Mass Balance

The spatial pattern of SMB differences between the higher-resolution ANTSI and the coarser-resolution AMIP (Fig. 3b) results from the interaction between the general atmospheric circulation and elevation differences resulting from enhanced spatial resolution (Fig. 1b). This results in lower SMB values in ANTSI (as compared to AMIP) towards the Ronne and Ross ice shelves and greater SMB values in the high interior. In particular, we note differences over the Antarctic Peninsula ("AP" in Fig. 3b), where SMB values are higher to the west and lower to the east, and West Antarctica ("WA" in Fig. 3b) where differences occur in regions strongly affected by orographic precipitation (discussed in Section 3.2).

[Figure 3]

To contextualize ANTSI AIS SMB results, we compare SMB results to those calculated from RACMO2.3p2 as well as the SMB reconstruction. (Fig. 3c,d)). Grounded AIS integrated SMB values for both AMIP and ANTSI are substantially higher than either RACMO2.3p2, at 1997 ± 92 GT $yr^{-1}$ or the reconstruction, at 1953 ±322 GT $yr^{-1}$, though there is substantial spatial variability in the differences. Over West Antarctica (denoted "WA" in Fig. 3b), both comparisons show similar bias patterns, whereby regions where ANTSI SMB estimates are lower show lower biases compared with the reference dataset, and where ANTSI SMB is higher, the bias is worse. As compared with RACMO2.3p2 results (where SMB values are available over ice shelves), results are mixed, with the Ronne, Amery and Larsen C ice shelves showing better agreement in ANTSI, but several other ice shelves (including the Ross ice shelf) showing lower biases with AMIP. Notably, over the Antarctic Peninsula, northern regions near the Larsen C ice shelf typically show lower biases with ANTSI results. While

large regions on Dronning Maud Land ("DM" 30°W to 60°W) towards the interior show a consistently lower bias in the coarser-resolution AMIP, other regions of East Antarctica show inconsistent bias increase/reduction between the two reference datasets.

### 3.1.3 Surface Mass Balance component: Surface melt

Mean spatially integrated surface melt values in ANTSI are similar to AMIP over both the Antarctic Peninsula and West Antarctica (i.e. the ANTSI mean ± one standard deviation of the temporal mean overlaps with the AMIP mean ± one standard deviation of the ensemble mean). Mean spatially integrated values for regions are shown in Supplemental Table 1 and temporal plots for spatially-integrated values are shown in Supplemental Fig. 2.

Over East Antarctica, values are significantly higher over ice shelves (ANTSI: 59 ±20 GT yr-1, AMIP: 30±4 GT yr-1), as well as the grounded ice sheet (ANTSI: 34 ±11 GT yr-1, AMIP: 11±2 GT yr-1). Trend analysis (where only significant trends, i.e. $p < 0.05$ are considered), indicates a weak positive trend in AMIP over East Antarctica (0.14 GT yr-2 over ice shelves, 0.07GT yr-2 over the grounded ice sheet) and a weak negative trend over the grounded ice sheet of West Antarctica (-0.09 GT yr-2). ANTSI shows a stronger positive trend over the ice shelves of the Antarctic Peninsula (0.26 GT yr-2). In summary, surface melt is intensified within ANTSI on some, but not all, ice shelves. These conflicting biases (despite all ice shelves being resolved to lower elevations in ANTSI) suggest that changes in surface melt are a product of both increased near-surface temperature as well as changes in atmospheric circulation (both of which can result from enhanced resolution). As compared to AMIP values, mean surface melt in ANTSI is enhanced over the Antarctic Peninsula on the western side, but reduced over the eastern side, including the Larsen C ice shelf; this is a direct impact of reduced westerly flow over the AP (discussed further in a later section focused on wind) resulting from the better resolved topographic barrier. Surface melt is enhanced over much of East Antarctica (Fig. 4a,b) and West Antarctica (Fig 4c,d) as well, including the Getz, Brunt Amery and Shackleton ice shelves (Fig. 5a). However, we note that over Amery and Shackleton ice shelves, this represents a correction of a negative bias within AMIP, while over most other ice shelves this represents an enhanced positive bias (Fig. 5b).

[Figure 4]
[Figure 5]

### 3.2 SMB Components: Precipitation

Precipitation is the dominant component of SMB over the AIS, and we find that spatial patterns of enhanced/reduced SMB (Fig. 3b) are driven by differences in large-scale precipitation, especially in winter (Fig. 6a), with the summer precipitation contribution to the surplus at higher elevations being smaller than those in winter (Fig. 6c) Over the larger Southern Ocean domain, precipitation is higher in ANTSI than in AMIP (potentially due to the higher resolution in ANTSI), with the

exception of the southern Weddell Sea, where the substantially better-resolved (higher) Antarctic Peninsula mountain range in ANTSI inhibits westerly flow and enhances the orographic precipitation gradient between its windward and leeward side. In addition, this enhanced precipitation is partially driven by enhanced meridional vapor transport towards the Weddell Sea sector and much of East Antarctica, especially in winter (Fig. 6b) followed by redistribution driven by enhanced zonal vapor transport in winter (a result of enhanced zonal windspeeds in ANTSI), noting that zonal vapor transport over the ocean is reduced in ANTSI overall (Supplemental Fig. 3). The atmospheric changes producing this enhanced poleward flow are discussed further in Section 3.3.

[Figure 6]

Enhancing resolution resulted in substantial changes to total cloud water path (TCWP, both liquid and ice). TCWP values aggregated by latitude are greater in ANTSI (as compared to AMIP) in winter, while in summer, ANTSI values are lower within the 75°S to 68°S latitude range. This net effect is a combination of changes in both the liquid and ice components, with the biases offsetting each other within the interior of Antarctica during summer. Total liquid cloud water path (TLCWP) is higher in ANTSI vs AMIP over the larger ocean domain during all seasons (Fig. 7c,d) and over the AIS during winter (Fig. 7c). As lower-resolution AMIP values already showed a high bias compared to ERA5, this represents a further-enhanced bias in ANTSI. However, TLCWP values are smaller in ANTSI over the interior grounded ice sheet during summer (Fig 6b) and this represents an improvement compared to ERA5 (Supplemental Fig. 3). Total cloud ice water path (TCIWP) is enhanced by increasing spatial resolution in all seasons, over the ice sheet as well as the interior. As AMIP simulations had a low bias compared to ERA5 estimates (Supplemental Figure 3c,i), this amounts to an improvement in bias over the summer (Fig. 7f), a reversal to a high bias over the interior ice sheet and lower-latitude Southern Ocean, and an improvement over the ocean immediately surrounding the AIS (Fig. 7e).

These results reflect similar spatial patterns in Greenland showing enhanced orographic precipitation in the Southeast and drying in the North (van Kampenhout et al., 2018) and we compare these to recent findings in the Discussion section (4).

[Figure 7]

## 3.3 Drivers of Enhanced Moisture Advection

The convergence of precipitation over the AIS is potentially a product of both moisture evaporated over the ocean and changes in winds that transport this moisture poleward. Cloud properties can be affected by the enhanced moisture in the atmosphere. As the Clausius-Clapeyron relationship specifies that water-holding capacity of air (saturation vapor pressure) increases 7% with every 1°C of warming (Held and Soden, 2006), even a small change in temperature can strongly affect moisture in the atmosphere and, potentially, cloud cover (though, in the model, the latter is a product of the specific

microphysical scheme). In this section, we discuss changes in temperature and wind structure, for both winter and summer, at the surface and aloft, as potential drivers for enhanced moisture advection towards the AIS.

### 3.3.1 Temperature

Overall, temperatures in ANTSI are higher than AMIP both at the surface and at higher levels in the atmosphere. Biases over the ice sheet are higher over the ice sheet than over the ocean, with the strongest biases shown during winter. Where biases are positive during winter over the ice sheet (the majority of the ice sheet), ANTSI biases are 1.76±1.26°C warmer than ERA5 and 2.62±1.27°C warmer than AMIP. Biases over the ice sheet during summer are positive over most of the ice sheet and slightly reduced in comparison to winter biases, with winter biases averaging 1.59±1.13°C as compared to ERA5 and 1.11±0.68°C as compared to AMIP. (Supplemental Table 2). Temperature biases at 500 hPa are substantially reduced overall (Supplemental Table 3).

Although temperatures are warmer overall, several regions are cooler, i.e. over the ocean at lower latitudes and to the east of the Antarctic Peninsula, including the Weddell Sea region (Fig. 8a,c). This pattern is likely driven by the enhanced blocking of the AP, which likely produce precipitation differences shown in Fig. 6a. As compared to ERA5 values in summer, the warmer atmosphere in ANTSI (Fig. 8b) reduces a cold bias present in AMIP (Supplemental Fig. **4o**), over the interior East Antarctic Ice Sheet and the ocean surrounding the AIS. However a warm bias in AMIP is slightly enhanced in ANTSI over the West Antarctic Ice Sheet and the regions of the East Antarctic Ice sheet nearest to it. Higher winter temperatures in ANTSI reduce a strong cool bias in AMIP (Supplemental Fig. 5o) as compared to ERA5 over both the ocean and ice sheet, even reversing signs to show a warm bias in portions of the East Antarctic Interior (Fig. 8d).

[Figure 8]

We also examine temperature biases as compared to AWS stations divided by season and elevation bins (Fig 9a-d), finding (a) a reduced positive temperature bias in ANTSI compared to larger positive biases in AMIP at elevations < 500m, but (b) a warm bias that intensifies with increasing elevation. This results in the partial correction of a cold bias at high elevation during the summer in ANTSI relative to AMIP (Fig. 9). In the winter and shoulder seasons within some higher-elevation areas, the bias changes in sign, from a cold bias in AMIP to a warm bias in ANTSI. We note that within shoulder seasons and winter, the warm bias in the > 3000m elevation is enhanced substantially in SON and JJA. AWS comparisons therefore demonstrate increasing biases with increasing elevation similar to patterns shown in the comparison with ERA5.

[Figure 9]

The warming produced by enhanced resolution is also shown at lower pressure levels, higher in the atmosphere. Comparisons of temperatures at 500 hPa in both summer (Supplemental Fig. 4p-r) and winter (Supplemental Fig. 5p-r) show enhanced warming over the entire domain, largely eliminating a cold bias in AMIP (compared to ERA5) over the AIS.

Previous work has discussed how this increase in temperature could be produced, namely by enhanced horizontal spatial resolution enhancing vertical velocities, thus generating greater condensational heating in CLUBB macrophysics

within CAM (Herrington and Reed, 2020). Overall, we consider the temperature changes produced by enhanced resolution to be an improvement.

### 3.3.2 Wind

The near-surface wind climate in Antarctica is governed by higher-speed katabatic winds and a large-scale pressure gradient force (PGF), both of which are stronger in winter, (van den Broeke and van Lipzig, 2003).

Seasonal differences in mean winter wind speed between AMIP vs ANTSI show enhanced wind speeds in ANTSI where higher elevations are resolved and lower wind speeds on the lee side of topography (e.g. the East Antarctic Peninsula). Generally, negative AMIP biases are improved, but not eliminated, in ANTSI in the interior as well as in the escarpment region, especially in winter (Fig. 10a,b). Where near-surface winter windspeed biases are positive (in comparison to the reference) over the ice sheet, mean ANTSI windspeeds are $1.00 \pm 1.13$ m/s higher than ERA5 and $0.53 \pm 0.55$ m/s higher than AMIP. All other mean biases over the ocean in winter or in summer for either the ocean or ice sheet differ by less than 0.8 m/s (Supplemental Table 3). We note that ANTSI near-surface wind speeds at higher elevations in East Antarctica are higher than AMIP (Fig. 10a), but lower than ERA5 (Fig. 10b). Comparisons between AMIP/ANTSI values and AWS stations, binned seasonally and by elevation, show negligible (< 1m/s) differences change in bias (Suppemental Fig. 6).

To better understand factors contributing to the strong convergence of water vapor over the region, we show a comparison between the meridional and zonal components at 500 hPa, finding that zonal wind speeds are reduced over the ocean, but enhanced in the interior (Fig. 10c) and that meridional wind speeds are higher from the ocean towards the center of East Antarctica. For near-surface as well as meridional and zonal wind speeds at 500 hPa, this represents an overall improvement as compared to AMIP simulations (Supplemental Fig. 5). These improvements are consistent (though less pronounced) in the summer season as well (Supplemental. Fig. 4).

[Figure 10]

In the aggregate, we find that both temperature and wind fields are improved by enhanced resolution, with the exception of excessive warming in the high interior during winter.

### 3.3.3 Disambiguating the Impacts of Dynamical Core vs Resolution

The ANTSI and AMIP-GOGA model configurations differ in both spatial resolution and dynamical core. To disambiguate whether the differences are a result of resolution or the dynamical core, we conducted a comparison between runs of two, 1° versions of CAM6 (One with the FV dynamical core; one with the SE dynamical core, both in a prescribed-SST configuration under historical radiative forcing for the 1979-1998 period. We call these the CAM6-FV and CAM6-SE runs, respectively.). Since the most notable differences between ANTSI and AMIP-GOGA are total cloud ice and liquid

water paths, we compare these variables in the CAM6-FV and CAM6-SE runs, finding no significant differences between them over Antarctica (Supplemental Fig. 7).

        Similarly, to assess the impacts of dynamical core and resolution on the atmospheric general circulation, we compare the Southern Annular Mode (SAM) and Pacific South American (PSA) patterns among all the model experiments and ERA5, using diagnostics code from the Climate Variability Diagnostics Package (Phillips et al., 2020). The SAM and

PSA patterns are widely recognized to be the leading modes of atmospheric circulation variability in the middle and high latitudes of the Southern Hemisphere on monthly to interannual timescales. Significant differences in these modes among the model configurations would have implications for the Antarctic surface mass balance and many other variables. For the AMIP-GOGA ensemble, we ran the diagnostics on each of the ten ensemble members. For the DJF SAM pattern (Supplemental Table 5), the spatial pattern correlations and RMS differences (compared with ERA5) from CAM6-SE,

CAM6-FV, and ANTSI are within the ensemble spread of AMIP-GOGA. In the monthly variability (Supplemental Figure 8), the SAM pattern in AMIP-GOGA #7 closely resembles the SAM pattern in CAM6-SE. The SAM pattern in AMIP-GOGA #5 resembles the SAM pattern in CAM6-FV as well as that in ANTSI. Such inter-ensemble spread in the SAM pattern is to be expected given the large internal variability in the atmospheric circulation. Similar results are obtained for the PSA-1 and PSA-2 patterns (Supplemental Figure 9), but somewhat complicated by the fact that PSA-1 and PSA-2 can

change positions with each other given their similar variance explained (Supplemental Table 5) and quadrature nature (e.g. Mo and Higgins, 1998). Overall, we do not find major circulation differences between CAM6-SE, CAM6-FV, and ANTSI, and, we cannot attribute these differences to the dynamical core or resolution. By extension, the subtle atmospheric circulation differences between ANTSI and AMIP-GOGA are not a good explanation for differences in their SMB or other variables; rather, these differences are likely to be driven by resolution.


### 3.4 The Surface Energy Budget

The energy available at the surface for melt is described below:

$QM = SW_{net} + LW_{net} + QH + QE + QR + GR$                                                            (4)

Here, $SW_{net}$ is net shortwave radiation ($SW_{net} = SW_{down} - SW_{up}$), $LW_{net}$ is net longwave radiation, ($LW_{down} - LW_{up}$), $QH =$ sensible heat transfer, $QE =$ latent heat transfer, $QR =$ energy supplied by rain, and $GR$ is $=$ energy supplied by ground flux. We have ignored QE, QR and GR for the purposes of this study, as we expect rain and ground flux to be minimal in
Antarctica (Hock et al., 2005).

In general, ANTSI shows increased $LW_{down}$ (with resultant impacts on the total energy balance), though this effect is reversed over the ice sheet in summer, and accompanied by greater $SW_{net}$. Generally, this comprises an improvement in biases as compared to ERA5. Although overall changes in the energy balance over the ice sheet are relatively small, the spatial patterns are coherent. All fluxes are defined positively when directed towards the surface, and mean seasonal values
for each element of the energy balance are provided as a reference for winter (Supplemental Figure. 10) and summer (Supplemental Figure 11. Additionally, full comparison of energy balance components comparing values to ERA5 are provided for winter (Supplemental Figure 12) and summer (Supplemental Figure 13).

### 3.4.1 Winter

In winter, when incoming shortwave radiation is negligible, net longwave radiation emitted from the surface is a
balance between incoming downward and outgoing cooling longwave radiation, with the deficit partially accounted for by changes to sensible heat flux. During winter, ANTSI shows an increase in $LW_{down}$ over the majority of the continent (Fig. 11d); where biases are positive over the ice sheet, mean winter ANTSI values are 6.78 $\pm$3.36 W m$^{-2}$ larger than AMIP (consistent with the warmer atmosphere in ANTSI)  However, $LW_{net}$ shows only a slight increase over East Antarctica and a decrease over much of West Antarctica. Over the ice sheet, positive biases for $LW_{net}$ average only 1.88 $\pm$ 1.57 W m$^{-2}$,
implying an increase in outgoing longwave radiation which produces enhanced surface temperatures shown in Fig. 8c.  This deficit is balanced partially by compensatory changes in sensible heat flux (Fig. 11a), with a mean positive bias compared to AMIP of 6.12 $\pm$ 9.98 W m$^{-2}$ and a mean negative bias of -2.75 $\pm$5.35 W m$^{-2}$.

Over the majority of the ocean domain in winter, ANTSI produces an increase in $LW_{down}$  (mean positive biases are 2.71 $\pm$ 2.67 W m$^{-2}$), $LW_{net}$  (mean positive biases are 2.06 $\pm$ 2.28 W m$^{-2}$) and sensible heat flux where positive values are
directed at the surface (mean positive biases are 3.22 $\pm$ 3.87 W m$^{-2}$, mean negative biases are -3.40 $\pm$ 5.53 W m$^{-2}$), where the Amundsen Sea sector and Weddell Sea sector show the reverse of the larger spatial pattern. Latent heat flux is reduced over the larger portion of the ocean (negative biases are -4.40 $\pm$ 5.95 W m$^{-2}$), corresponding to greater latent heat flux from the ocean towards the atmosphere, excepting sections of the Indian Ocean at lower latitudes.

Because SSTs and SICs were forced from observations in this model run, we assume that this behaviour is the result
of changes in wind patterns and temperature produced by enhanced resolution over the ocean domain. As compared to ERA5 reanalysis, the bias changes in energy balance components in winter are relatively minor except for downward longwave radiation, where the enhanced resolution produces greater agreement with ERA5 over East Antarctica; the mean negative

bias over the ice sheet, as compared to ERA5, decreases from $-9.38 \pm 4.91$ W m$^{-2}$ (AMIP) to $-4.56 \pm 2.56$ W m$^{-2}$ (ANTSI).
The only degradation in winter is a slight increase in mean positive longwave radiation biases (Table 2). We note, in light of
the total cloud liquid water path in ANTSI showing greater disagreement with ERA5, that downward longwave radiation is a
product of many factors other than cloud liquid water path. In the aggregate, we conclude that the radiation balance in winter
is improved with enhanced spatial resolution.

[Table 2]

[Figure 11]

### 3.4.2 Summer

In summer (DJF), enhanced LW$_{down}$ in ANTSI is also greater than in AMIP, though the bias is muted as compared to winter,
extending over most of the ocean domain except for the Weddell Sea sector and over East Antarctica. Mean summer biases
(ANTSI – AMIP) are less than 4 W m$^{-2}$ over the ocean as well as over the ice sheet. Over the margins of the ice sheet and
West Antarctica, LW$_{down}$ is reduced, partially matching the pattern of reduced total liquid cloud path (Fig. 7b). Over most of
the ocean, net and downward longwave radiation are both positive, implying that LW$_{down}$ is not entirely balanced by
longwave cooling (LW$_{up}$)

Over the majority of the ice sheet, we show a reduction in LW$_{net}$, as LW$_{up}$ exceeds the surplus LW$_{down}$ . LW$_{down}$ is
reduced over the ocean (nearly by 20 W m-2 in regions of the Indian and Pacific sector) and over the center of East
Antarctica, but greatly increased over the margins as well as the Weddell Sea sector (nearly 15 W m$^{-2}$ in some regions). The
latter is consistent with cloud clearing due to the higher topographic barrier. SW$_{net}$ decreased substantially over the ocean
and increased over the ice sheet.

In comparison to ERA5, the biases over oceans for LW$_{down}$, LW$_{net}$, SW$_{down}$ and SW$_{net}$ increase in ANTSI (Fig. 12,
Table 3, Supplemental Figures 12 and 13). Over the ice sheet, biases show a distinct spatial pattern, with the bias sign
reversing between the high plateau in East Antarctica vs West Antarctica and the margins of the ice sheet. In the aggregate,
the radiation balance is improved or changed only slightly, with mean biases reduced (Table 3) and biases within ANTSI
falling within one standard deviation of the ERA5 mean (Supplemental Fig. 13h,I,n,o,q,r). The exception is SW$_{net}$
(generally a positive bias), where ANTSI biases are higher overall.

In summary, during summer, the representation of the radiation balance show reduced biases as compared to ERA5
over the ice sheet (with the exception of net shortwave radiation), but over the ocean, biases in both shortwave and longwave
radiation have increased within ANTSI. One exception to this generalized spatial pattern is the Weddell Sea sector, where
the better-resolved (higher) Antarctic Peninsula blocks westerly flow.

[Table 3]

[Figure 12]

## 4. Discussion and Conclusions:

In this study, we have compared the climatology in the standard-resolution, 1° CESM2 with the variable-resolution CESM2, which has high resolution (as fine as 0.25°) over Antarctica and the Southern Ocean. We find that enhanced horizontal resolution generally reduces climatological biases, except with regard to the formation of clouds over the Southern Ocean. Temperature and wind speed biases (both near-surface and aloft) are largely reduced over both the ocean and the ice sheet, and surface mass balance estimates are generally equivalent, except for over the Antarctic Peninsula. Here, the high resolution resolves the spine of the Antarctic Peninsula mountain range to over 400m greater height, resulting in enhanced precipitation on the western side and reduced precipitation, lower temperature and decreased surface melt on the eastern side and the Weddell Sea sector. In comparison to an ice-core-based, gridded reconstruction as well as the regional climate model RACMO2.3p2, SMB biases from both datasets produce similar spatial patterns for changes in bias over the AP, West Antarctica and some regions of East Antarctica. There is better agreement between the enhanced-resolution model (ANTSI) and the reference datasets over lower-elevation regions in West Antarctica and sections of East Antarctic nearest to the Ronne-Filchner and Ross ice shelves, while biases have increased over Dronning Maud Land. These changes are primarily driven by greater precipitation in these regions driven by enhanced meridional vapor transport from the oceans, especially in winter.

While temperature, wind (both zonal and meridional), specific humidity and total cloud ice water path over the ocean are improved, biases are increased for total cloud liquid water path (TCLWP) over the ocean region in both summer and winter, with a concurrent decrease in downward and net shortwave and increase in downward and net longwave in summer. Over the ice sheet, TCLWP biases are higher in winter as compared to ERA5, but the increase in downward longwave radiation (partially affected by TCLWP) amounts to a reduction in bias compared to ERA5. By contrast, summer TCLWP is higher at the margins, but lower in the interior of the ice sheet, a pattern conforming to changes in orographic precipitation produced by enhanced resolution reported by Kampenhout (2019). The summer pattern for biases in downward and net longwave and downward shortwave (but not net shortwave) indicate reduced biases. The change in downward shortwave radiation and higher temperatures may cause another bias, namely the increase in surface melt over most ice shelves, excepting the eastern side of the Antarctic Peninsula (where temperatures are kept lower by the orographic barrier). We note that only for a few ice shelves (such as the Brunt and interior of the Ross Ice Shelf) does this represent a significant relative increase, as compared to the already-high bias between the coarser-resolution AMIP and observations (QuikSCAT). The increased surface melt also represents an improvement in bias compared to QuikSCAT over the Amery and Shackleton ice shelves, where coarser-resolution simulations underpredict surface melt.

Biases are largely reduced in the near-surface atmosphere over the ice sheet and ocean. However, the physics of cloud formation (and resultant precipitation), especially over the ocean, may be better paramaterized for the coarser resolution model. Future improvements to CAM may benefit from examining how paramaterizations affect cloud formation over the Southern Ocean in particular, where there are few land masses and where the enhanced availability of precipitation

can strongly impact surface mass balance over the ice sheet. Currently, tuning for CESM2 has focused on the standard 1° grid, while future parameterizations will account for the effects on variable-resolution grids (including ANTSI). It is also likely that more detailed evaluations of clouds and related properties, employing different observational datasets and cloud simulators, will help to better pinpoint deficiencies in all versions of the model than we have been able to do here with the reanalysis. We also note the possibility that these biases could be alleviated with the activation of the high-resolution ocean component of CESM2, rather than the one-way forced sea surface temperature and sea ice concentration formulation used in these simulations. Notably, recent work employing a global, fully coupled high-resolution (0.25° atmosphere; 0.10° ocean) decadal prediction system with CESM1 suggests that high-resolution over the Southern Ocean in particular is key for improving climate simulations across the globe (Yeager et al., 2023). However, that work was extremely computationally expensive; a coupled implementation of VR-CESM2 with enhanced resolution over the ANTSI domain might provide similar improvements at much lower expense.

By providing knowledge of its strengths and limitations, this work illustrates that the variable-resolution configuration over Antarctica is a valuable tool for representations of precipitation, surface mass balance, and therefore sea level rise estimates. In particular, the enhanced resolution better articulates spatial patterns associated with orographic precipitation, which may capture fine-scale changes in precipitation patterns in future warming scenarios. We note that the improvements over Antarctica are less pronounced than those over Greenland, where they are significant (van Kampenhout et al., 2019). The enhanced resolution grid captures these changes while preserving two-way interaction with the global climate system, but at a reduced computational cost compared to global high-resolution runs. However, we note that some regional climate models such as RACMO2.3p2 (van Wessem et al., 2018) or the MAR model (Agosta et al., 2019) contain more detailed representations of the firn layer as well as more complex albedo schemes. Therefore, RCMs and VR-ESMs currently serve different purposes. For studies specifically focused on the effects on the two-way interactions with firn, RCMs such as RACMO2.3p2 or MAR may provide better value overall. For these ANTSI runs, we note that outputs have been produced at 3-hourly intervals that could be used to force offline firn models with even finer representation of firn layers than RCMs. Similarly, despite a poorer representation of the firn layer, very high-resolution non-hydrostatic models (Gilbert et al., 2022) best capture wind and temperature dynamics over the East Antarctic Peninsula, e.g. foehn flow. While the variable-resolution domain provides substantial value for experiments fully utilizing the two-way interaction between the interior and global domain, they do not replace RCMs.

Future work will take advantage of the enhanced resolution and relatively low computational cost of ANTSI. While our evaluation of ANTSI has focused on the 1979-2014 period (to match available AMIP results), results from ANTSI are available for the 2015-2020 period, during which we have implemented moisture-tagging, connecting precipitation to its sources. As infrequent and sometimes small spatial scale extreme events constitute a substantial fraction of mean-annual precipitation over much of the Antarctic Ice Sheet (Dalaiden et al., 2020), high resolution will offer better insights into their dynamics and impacts. In future work, we will use ANTSI to present the climatology and moisture sources of atmospheric rivers over the Antarctic Ice Sheet. We also anticipate the use of the ANTSI grid in fully coupled runs for future climate

scenarios, better constraining the dynamics of precipitation over the Antarctic Ice Sheet and thus future estimates of sea level rise.

**Data availability:**

All outputs used for this evaluation are available through Zenodo: https://doi.org/10.5281/zenodo.7335892

**Author contribution:**

RTD performed simulations and major analysis. AH provided substantial guidance about the usage of VRCESM2. JTML provided initial guidance on the project design. DPS provided input on the analysis and on the final version of the manuscript. ZY and LT provided input on the final manuscript. DD provided input on initial versions of the manuscripts.

**Competing Interests:**

The authors declare that they have no conflict of interest

**Acknowledgements:**

Funding for this work, through RTD, JTML, DPS, DD and ZY was provided by grant 1952199 from the National Science Foundation (NSF), Office of Polar Programs. RTD and LT also received funding through NASA Grant S000885; DPS was 520 also supported through the National Center for Atmospheric Research, which is a major facility sponsored by the NSF under Cooperative Agreement no. 1852977. The authors thank all the scientists, software engineers, and administrators who contributed to the development of CESM, which is primarily supported by NSF. Finally, the authors appreciate the efforts of the reviewers and editors, whose constructive comments led to an improved manuscript.

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

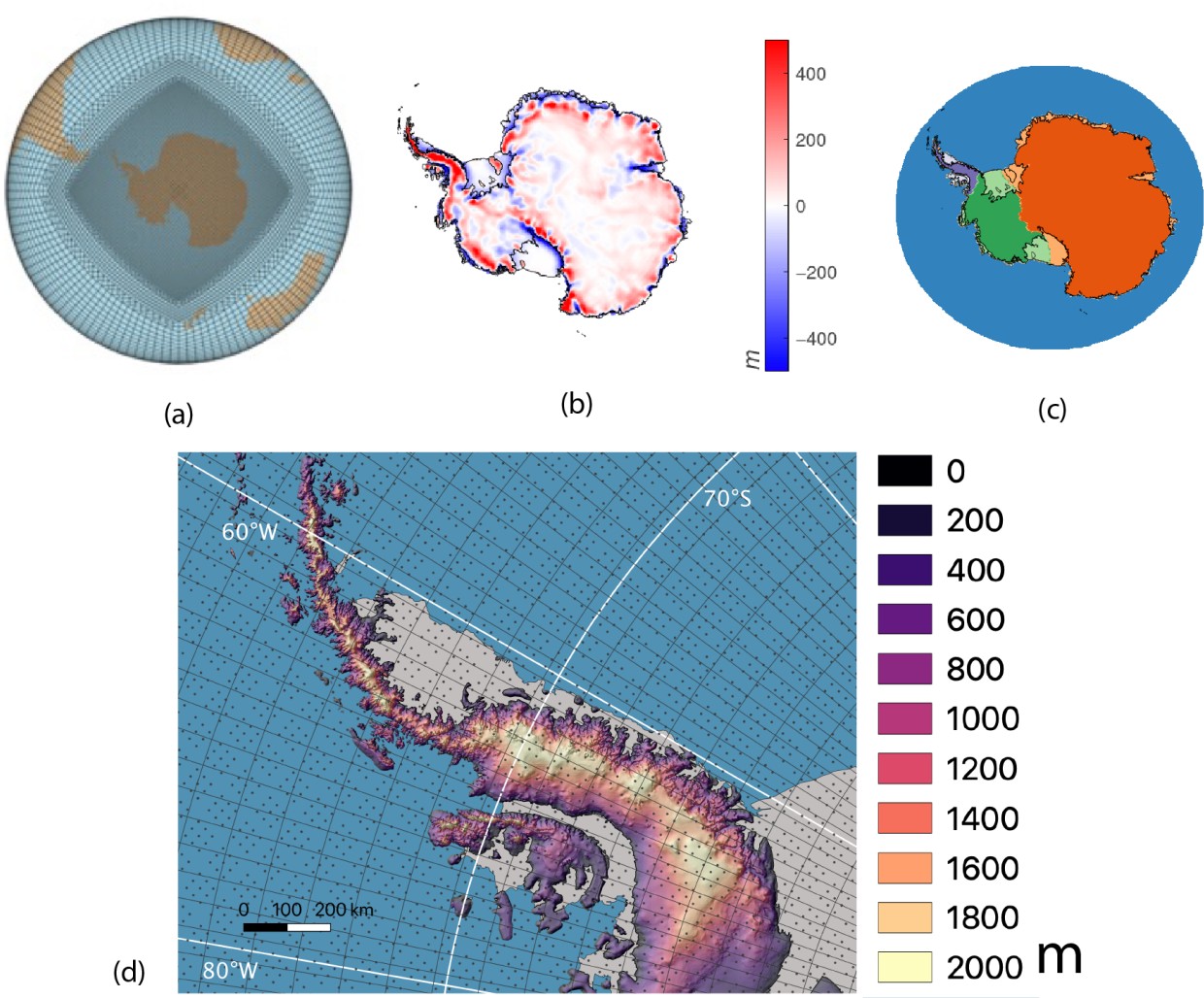

(a)

(b)

(c)

(d)

 **Figure 1: Elevation (a) the ANTSI grid with 0.25° spectral element points in the interior and 1° in the exterior (b) Difference between interpolated surface elevation (ANTSI-AMIP) (c) regions shown for East Antarctica (orange), West Antarctica( green) and the Antarctic Peninsula (purple) with lighter shades indicating ice shelves and darker shades indicating the grounded ice sheet, from Zwally et al., 2002 (d) Elevation (source: ETOPO), AMIP grid shown in lines, ANTSI spectral element points shown in dots over the Antarctic Peninsula**

| Region | Full | | Grounded Ice Sheet | | Ice Shelves | |
|---|---|---|---|---|---|---|
| model | AMIP | ANTSI | AMIP | ANTSI | AMIP | ANTSI |
| East. Antarctica | 1492.93 (± 43.57) | 1583.66 (±101.93) | 1283.83 (± 38.26) | 1394.32 (± 91.25) | 209.10 (±7.29) | 189.35 (± 17.86)<br><br>-0.80 GT yr$^{-2}$ |
| West Antarctica | 909.47 (±41.30) | 933.49 (±72.34) | 696.88 (±33.98) | 727.66 (±59.74) | 212.59 (±8.51) | 205.84 (±15.08) |
| Antarctic Peninsula | 112.99 (±6.40)<br><br>0.29 GT yr$^{-2}$ | 151.17 (±13.28)<br><br>0.56 GT yr$^{-2}$ | 71.81 (± 4.34)<br><br>0.24 GT yr$^{-2}$ | 106.38 (±10.48)<br><br>0.51 GT yr$^{-2}$ | 41.18 (±2.49) | 44.79 (±5.10) |
| ALL | 2515.39 (±53.14) | 2668.32 (±123.10) | 2052.2 (±44.89) | 2228.35 (±111.77) | 462.87 (±11.32) | 439.97 (±20.56) |


**Table 1: Surface mass balance values (in GT/yr) for each region, model source as listed. Interannual trends shown in red using the Mann-Kendall test where p values are < 0.05**

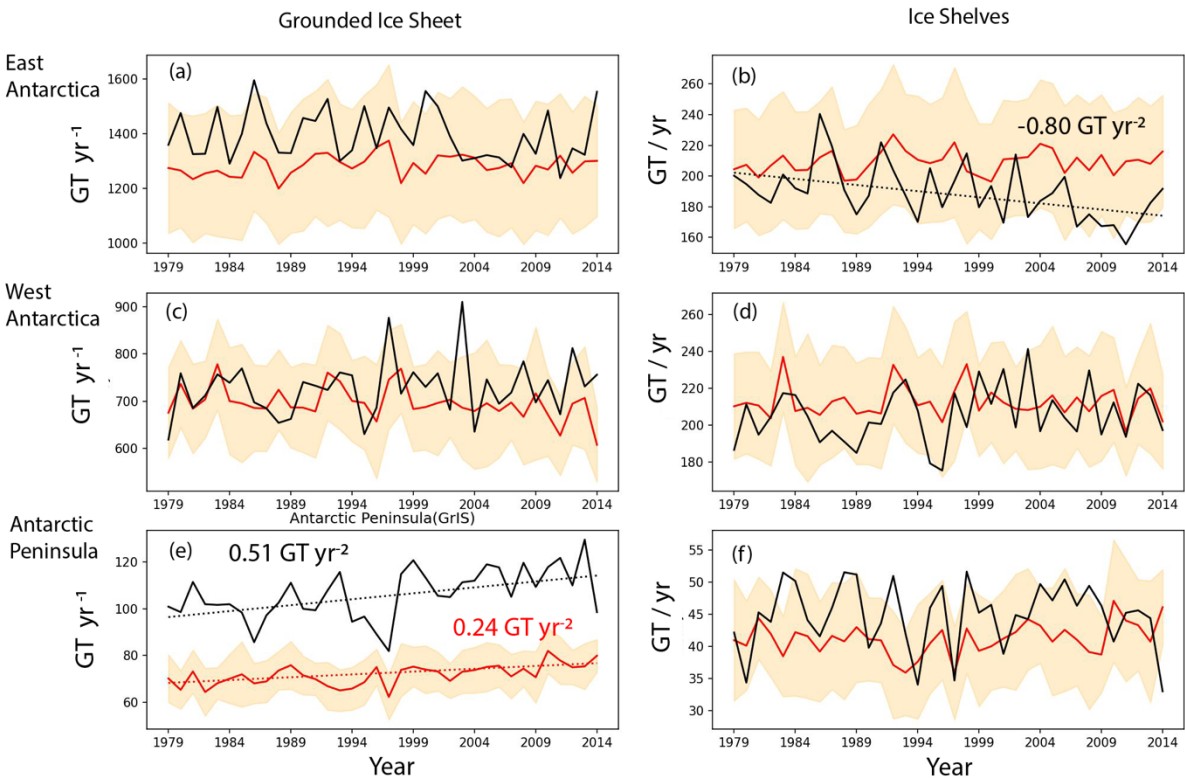

**Figure. 2 Integrated surface mass balance for AMIP (mean shown in orange line, one standard deviation shown in orange shade) vs ANTSI (black line) for (a) East Antarctic Grounded Ice Sheet (b) East Antarctic Ice Shelves (c) West Antarctic Grounded Ice Sheet (d) West Antarctic Ice Shelves (e) Antarctic Peninsula Grounded Ice Sheet (f) Antarctic Peninsula Ice Shelves. Trend lines shown where the Mann-Kendall trend test  p-value  < 0.05**



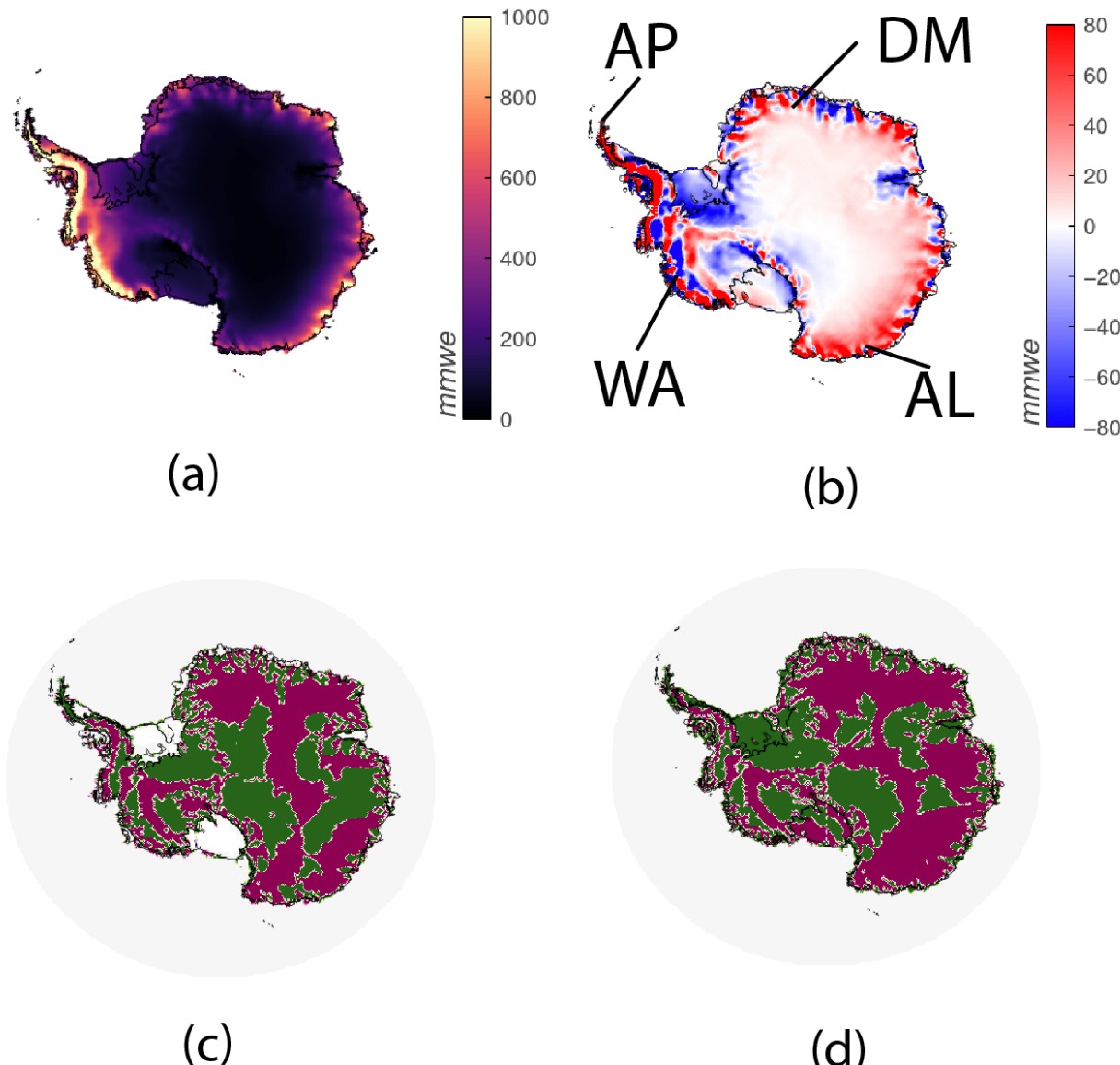

**Figure 3. Surface mass balance (SMB) over the 1979-2014 period. (a) SMB from ANTSI (b) difference in SMB (ANTSI-AMIP). Regions indicated include the Antarctic Peninsula (AP), Dronning Maud Land (DM), West Antarctica (WA) and Adelie Land (AL). Panels (c) and (d) show ANTSI SMB bias relative to AMIP compared to the Reconstruction and RACM2.3, with green and purple indicating reduced/increased bias in ANTSI relative to the dataset**

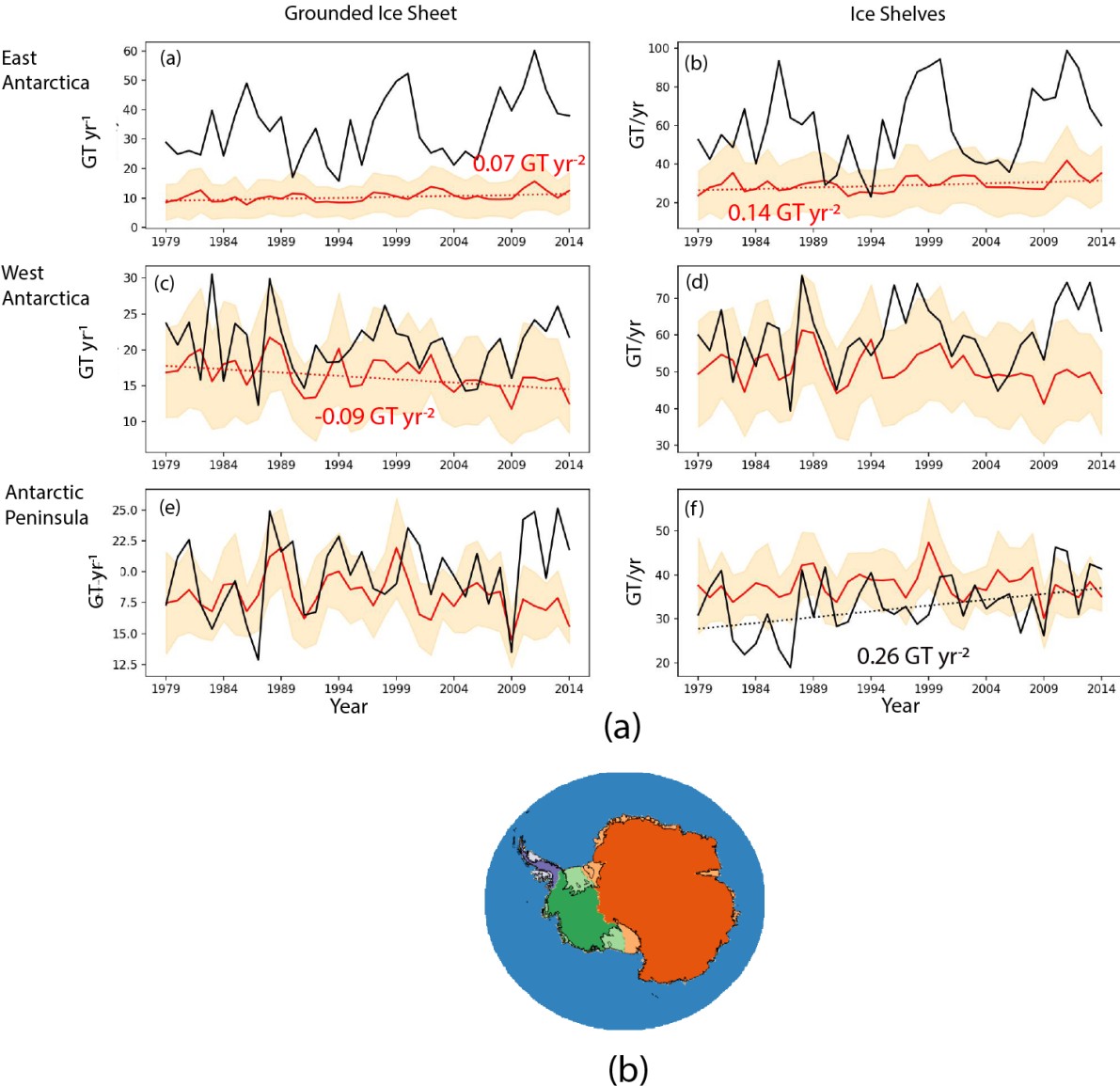

**Figure 4: (a) Surface melt (in GT yr-1) shown (b) Regions for East Antarctica (orange), West Antarctica (green) and the Antarctica Peninsula (purple), with darker colors indicating the grounded ice sheet and lighter colors indicating ice shelves. Based on Zwally et al. 2002**

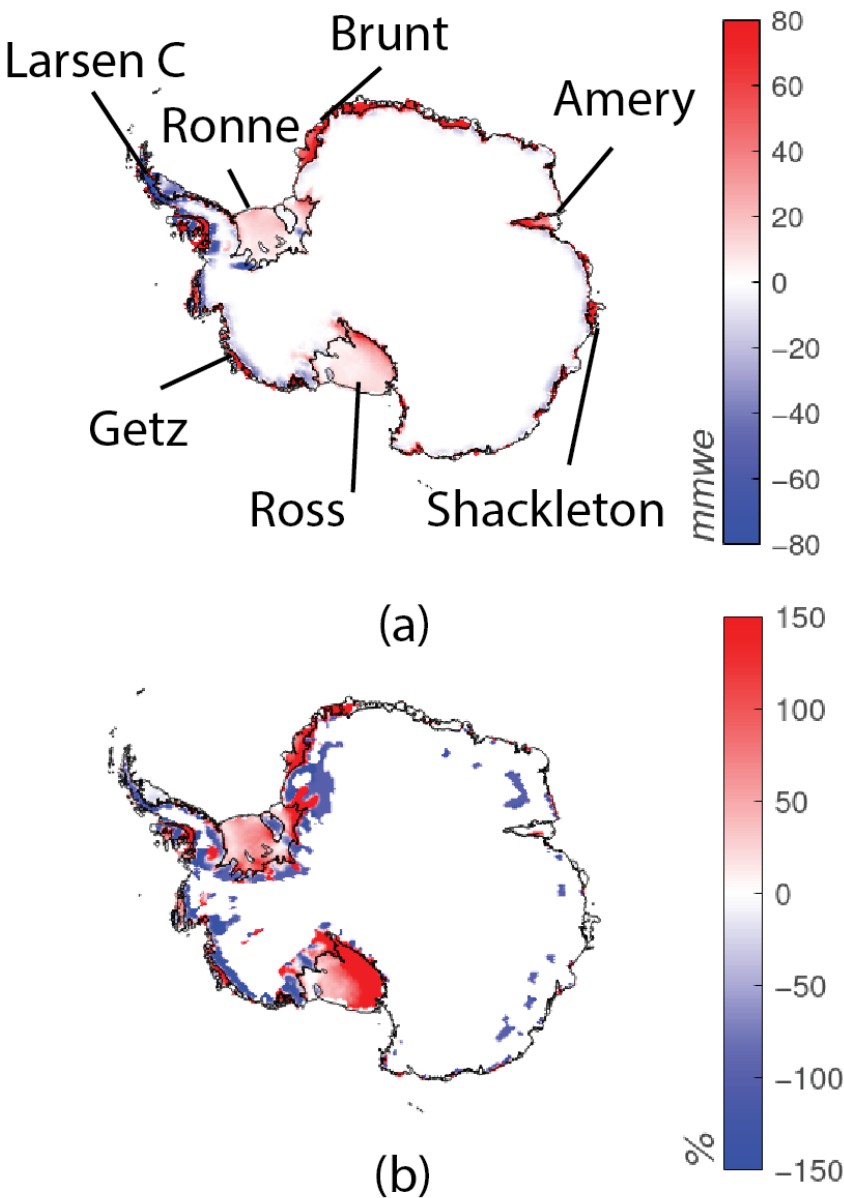


**Figure 5:** **Changes in surface melt with resolution. (a) Mean difference in surface melt (ANTSI − AMIP), with individual ice shelves indicated (b) Restricted to where AMIP surface melt shows a positive bias as compared to QScat, with red values indicating higher biases. This is the relative difference between models as a percentage, i.e. (ANTSI − AMIP)/(AMIP-QSCAT) * 100.**

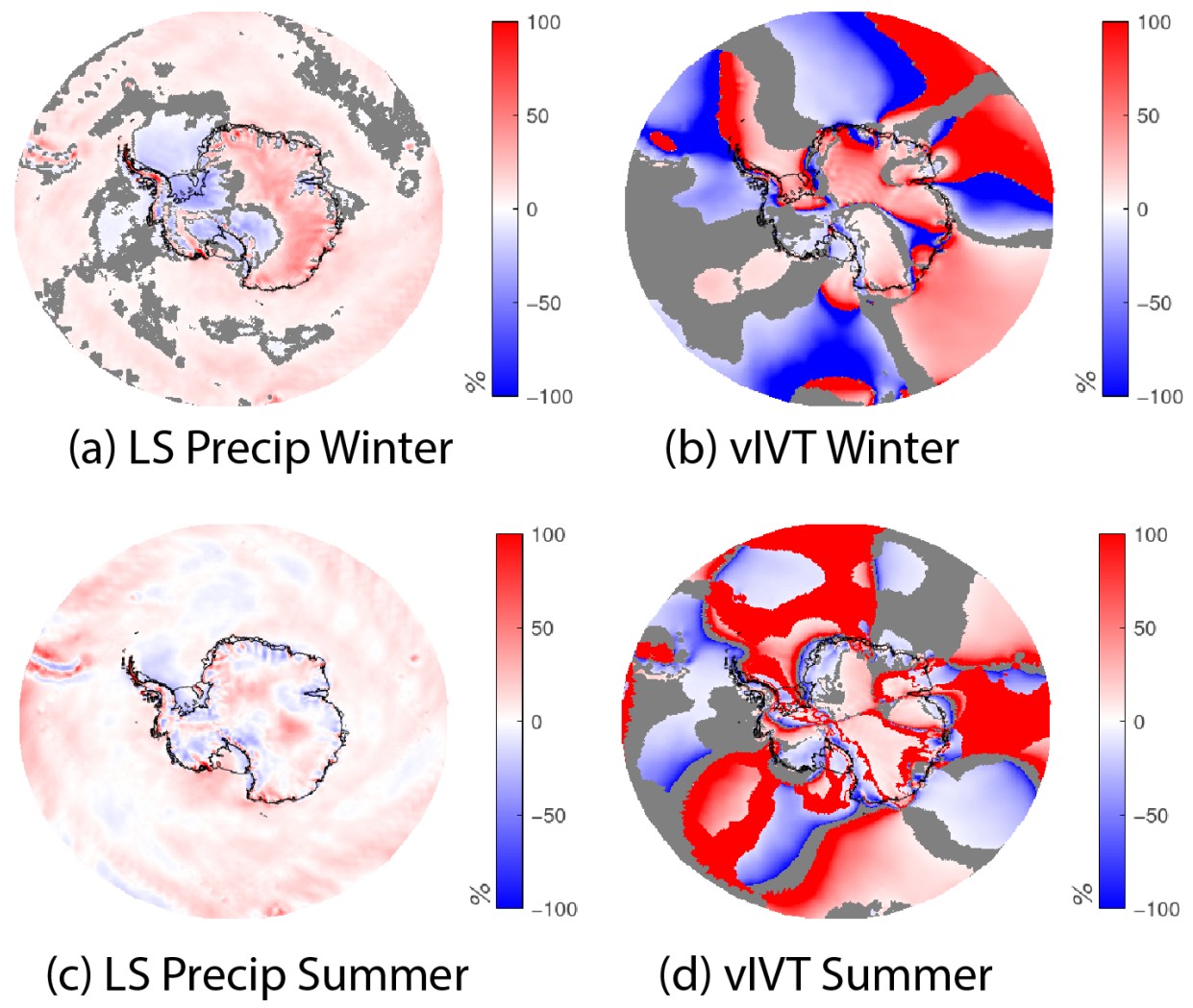

**Figure 6: Relative change (%) in large-scale (LS) precipitation and integrated meridional vapor transport (vIVT) between ANTSI and AMIP, 1979-2014 seasonal mean (grey indicates where biases are not significant as compared to AMIP ensemble std dev). (a) large-scale precipitation for winter (JJA) and (c) summer (DJF). Integrated meridional vapor transport for winter (b) and summer (d)**


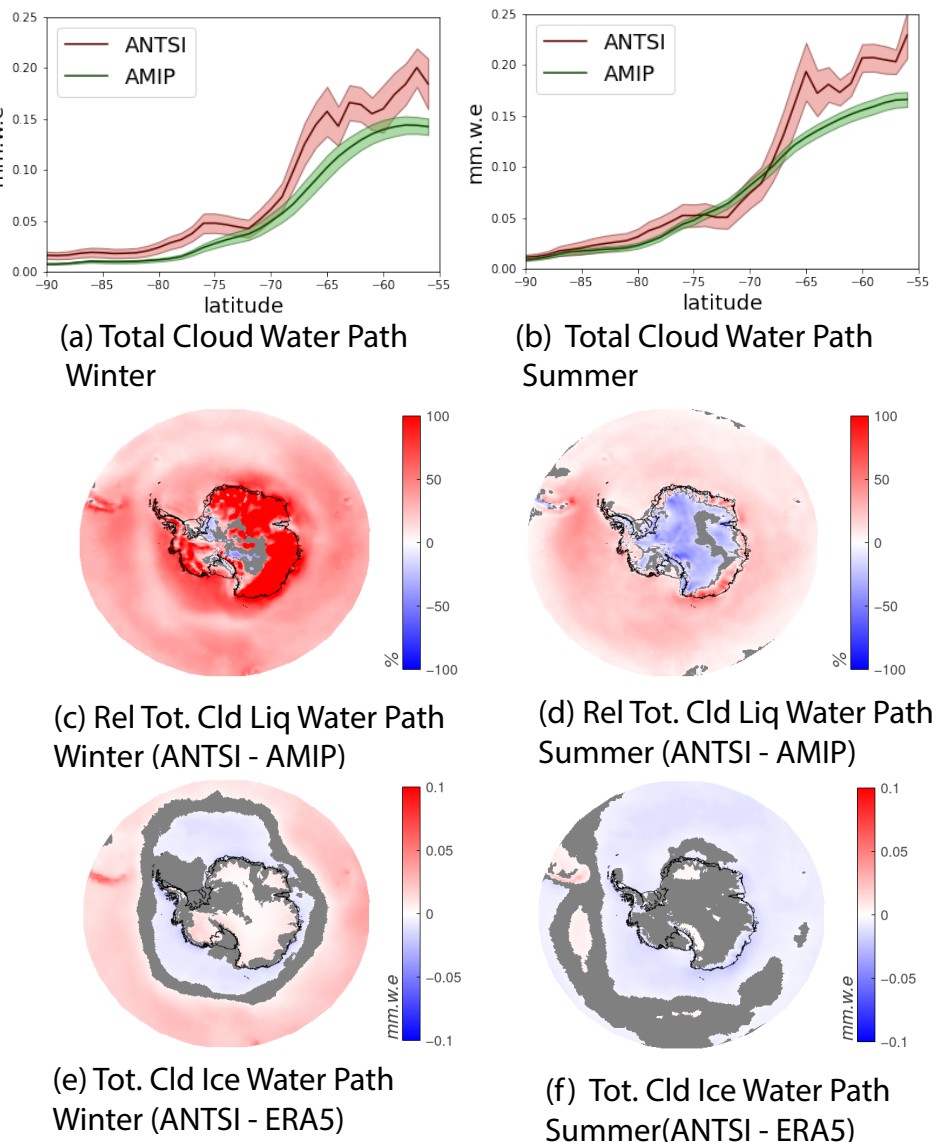

(a) Total Cloud Water Path Winter

(b) Total Cloud Water Path Summer

(c) Rel Tot. Cld Liq Water Path Winter (ANTSI - AMIP)

(d) Rel Tot. Cld Liq Water Path Summer (ANTSI - AMIP)

(e) Tot. Cld Ice Water Path Winter (ANTSI - ERA5)

(f) Tot. Cld Ice Water Path Summer(ANTSI - ERA5)


**Figure 7: Cloud water path. Total cloud water path aggregated in latitude bands (shaded area shows temporal standard deviation) for (a) winter, JJA (b) Summer, DJF. Relative difference between ANTSI and AMIP (in percentage) in total cloud liquid water path for seasonal mean (1979-2014) for winter, JJA (c) and summer, DJF (d). Absolute difference between ANTSI and ERA5 reanalysis in total cloud ice water path for seasonal mean (1979-2014) for winter, JJA (e) and summer, DJF (f). Grey indicates**
**where biases are not significant as compared to AMIP ensemble std dev.**

## (a) DJF (ANTSI - AMIP)

## (b) DJF (ANTSI - ERA5)

## (c) JJA (ANTSI - AMIP)

## (d) JJA (ANTSI - ERA5)

**Figure 8. Near surface temperature comparisons between models and reanalysis. (a) DJF ANTSI-AMIP (b) DJF ANTSI - ERA5 (c) JJA ANTSI-AMIP (d) JJA ANTSI-ERA5 where difference exceeds one standard deviation of AMIP ensemble mean. Grey indicates where biases are not significant as compared to AMIP ensemble std dev.**

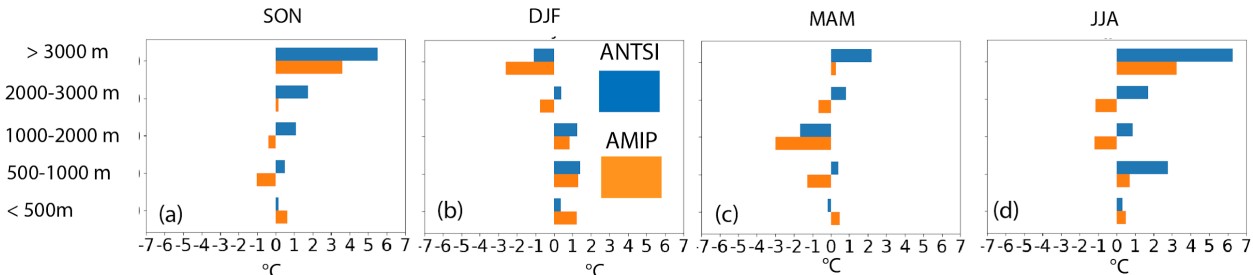

**Figure 9: Near-surface temperature comparisons with AWS stations. Temperature biases as compared to AWS stations, binned into elevation classes (a) SON (b) DJF (c) MAM (d) JJA**

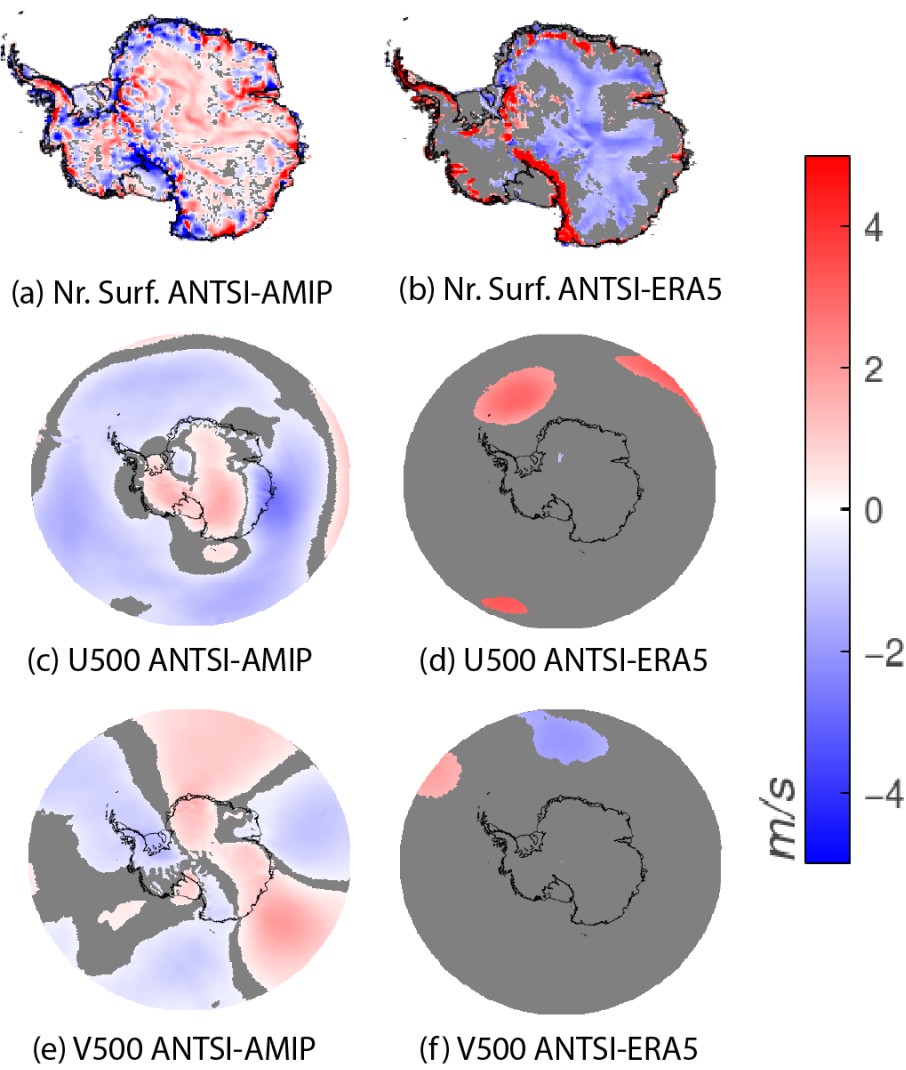

(a) Nr. Surf. ANTSI-AMIP  (b) Nr. Surf. ANTSI-ERA5

(c) U500 ANTSI-AMIP  (d) U500 ANTSI-ERA5

(e) V500 ANTSI-AMIP  (f) V500 ANTSI-ERA5


**Figure 10. Mean winter (JJA) wind speed differences (1979-2014) between ANTSI-AMIP (left column) and ANTSI-ERA5 (right column) for the near-surface (a)(b), the 500 hPa zonal component (c)(d) and the 500 hPa meridional component (e)(f).** Grey indicates where biases are not significant as compared to AMIP ensemble std dev.


| | Ocean | | Ice sheet | |
|---|---|---|---|---|
| | $LW_{down}$ | $LW_{net}$ | $LW_{down}$ | $LW_{net}$ |
| **AMIP – ERA5** | | | | |
| Positive bias | 10.72 ±6.23 | 10.29 ± 5.19 | 5.11 ± 5.18 | 5.22 ± 4.44 |
| Negative bias | -4.29 ±5.53 | -7.70 ± 7.81 | -9.38 ± 4.91 | -6.55 ± 5.31 |
| **ANTSI – ERA5** | | | | |
| Positive bias | **10.78 ± 6.34** | **10.51 ± 5.52** | **4.60 ± 3.67** | **4.25 ± 4.04** |
| Negative bias | **-4.27 ± 5.50** | **--6.38 ± 6.79** | **-4.56 ± 2.56** | **-6.16 ± 5.44** |

**Table 2: Mean and standard deviation of downward and net longwave radiation (model - ERA5) in winter (JJA). Values are calculated separately for the sign of bias (positive and negative) and for the ocean (below -55° latitude) vs. the ice sheet. Bias improvement is shown in green and degradation in red.**

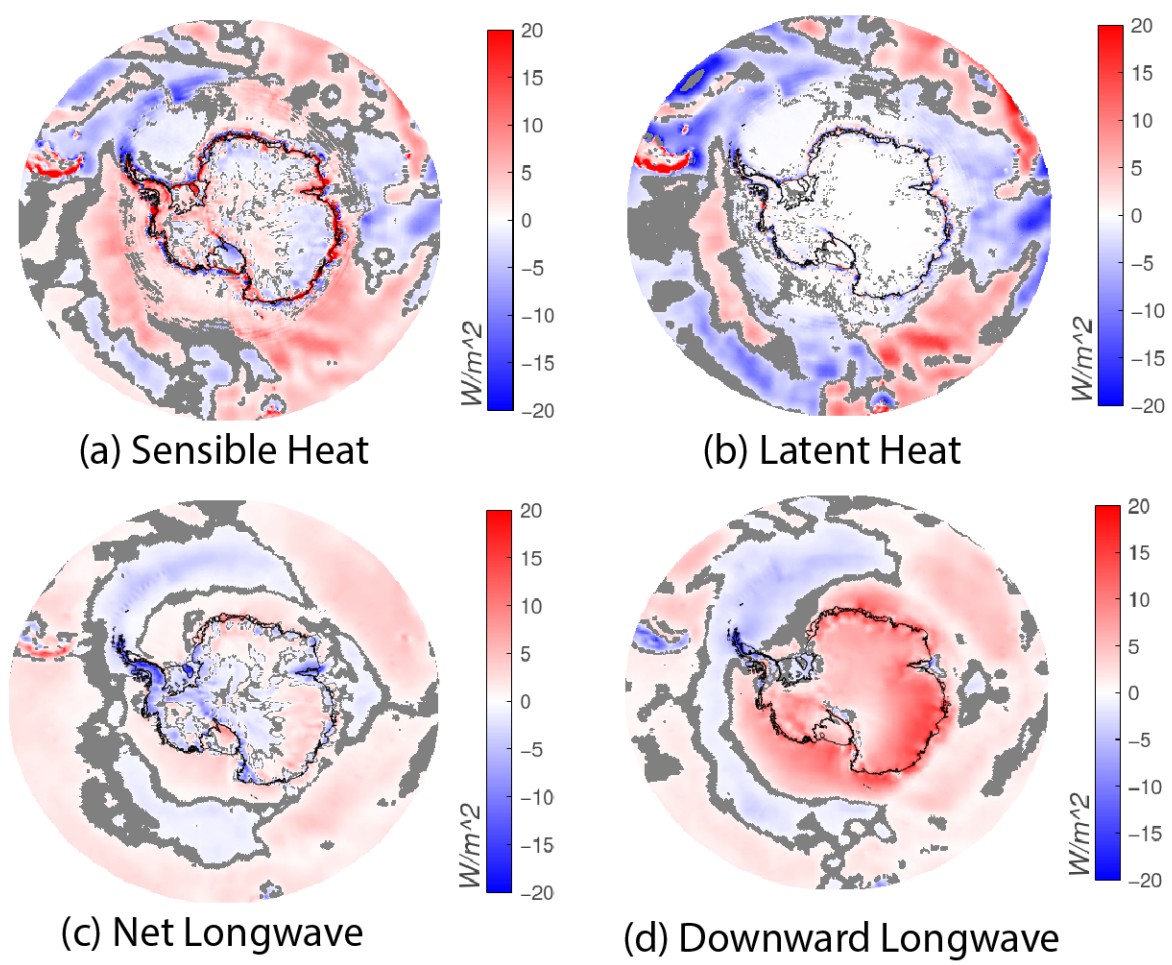

**Figure 11: Mean differences (ANTSI-AMIP) in the winter (JJA) radiation balance (1979-2014) for (a) Sensible Heat Flux (b) Latent Heat Flux (c) Net Longwave Radiation (d) Downward Longwave Radiation. All fluxes are directed downward.**



| | Ocean | | | | Ice sheet | | | |
|---|---|---|---|---|---|---|---|---|
| | $LW_{down}$ | $LW_{net}$ | $SW_{down}$ | $SW_{net}$ | $LW_{down}$ | $LW_{net}$ | $SW_{down}$ | $SW_{net}$ |
| **AMIP-ERA5** | | | | | | | | |
| Positive bias | 12.63 ±6.21 | 12.08 ±5.95 | 13.21 ± 10.47 | 13.72 ± 10.69 | 12.59 ± 8.49 | 7.85 ± 6.31 | 8.91 ± 5.96 | 6.60 ± 4.88 |
| Negative bias | -5.75 ±4.95 | -6.83 ±6.24 | -28.36 ±17.74 | -27.25 ±16.58 | -5.96 ± 3.69 | -4.21 ± 2.77 | -14.72 ± 11.28 | -20.18 ± 24.80 |
| | | | | | | | | |
| **ANTSI-ERA5** | | | | | | | | |
| Positive bias | **13.67 ± 6.89** | **12.89 ± 6.28** | **12.12 ±10.70** | **12.87 ± 10.76** | **11.32 ±6.71** | **5.90 ± 4.81** | **7.42 ± 4.95** | **8.13 ± 5.79** |
| Negative bias | **-4.83 ± 4.95** | **-5.82 ± 5.52** | **-33.00 ± 20.08** | **-32.04 ± 1964** | **-3.50 ± 2.20** | **-4.27 ±3.05** | **-11.82 ± 9.34** | **-23.66 ± 24.98** |

**Table 3: Mean and standard deviation of downward and net longwav eand shortwave radiation (model - ERA5) in winter (DJF). Values are calculated separately for the sign of bias (positive and negative) and for the ocean (below -55° latitude) vs. the ice sheet. Bias improvement is shown in green and degradation in red.**

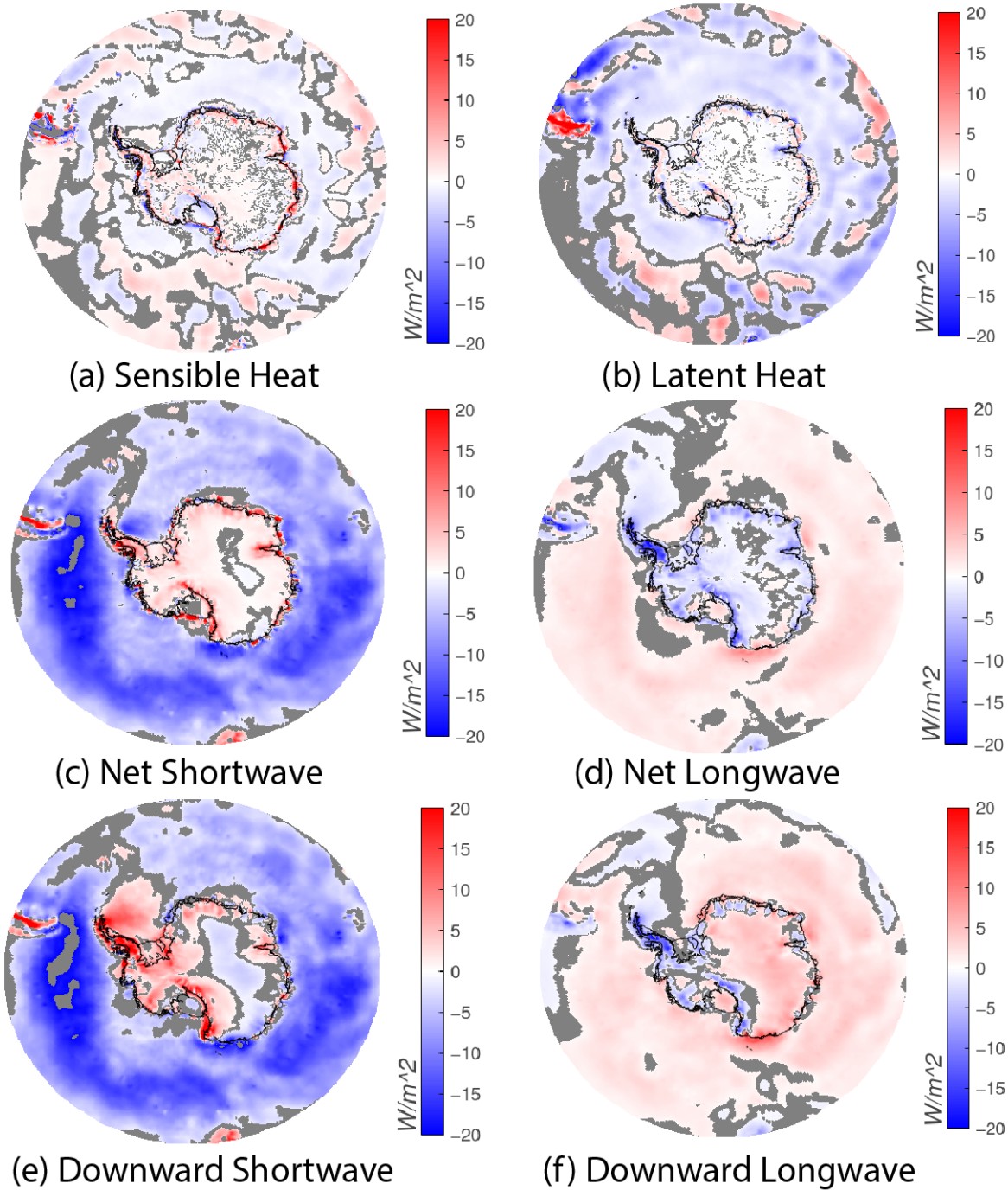

**Figure 12: Mean differences (ANTSI-AMIP) in the summer (DJF) radiation balance (1979-2014) for (a) sensible heat flux (b) latent heat flux (c) net shortwave radiation (d) net longwave radiation (e) downward shortwave radiation, (f) downward longwave radiation. All fluxes are directed downward.**