# Peer review of "Evaluating the Impact of Enhanced Horizontal Resolution over the Antarctic Domain Using a Variable-Resolution Earth System Model"

_EGUsphere, 2022_

## Referee Comment (RC1)

**Review of "Evaluating the Impact of Enhanced Horizontal Resolution over the Antarctic Domain Using a Variable-Resolution Earth Systems Model" by R. T. Datta et al., submitted to *The Cryosphere***

**General comments**

The manuscript presents a systematic evaluation of the impact of using a high-resolution nest over the Antarctic within the CESM variable resolution (VR) system. The authors find that there is a general improvement associated with increasing the resolution over Antarctica when compared with AMIP and against other observational datasets. They note that the VR configuration represents a compromise between RCMs (which can fail to capture the influence of the global climate) and high-res GCMs that are often impractical to run.

It is very well presented, with suitable figures, tables and supplementary materials. The text is logically structured and contains sound arguments. The use of VR modelling is a growing area of research and will as such be of interest to readers in the cryospheric climate sciences. Although the manuscript is particularly focused on the evaluation of VR-CESM2, some of the findings may be relevant to researchers beyond the CESM2 community. While it does not introduce new concepts or results, the manuscript is a useful benchmark and an important contribution to the community.

I recommend the paper be accepted, subject to minor revisions. A few minor modifications and clarifications in the presentation of the manuscript would improve its usefulness to readers, and I have detailed these below.

**Specific comments**

**Main text**

Have you tested different VR nests? e.g. 0.1 deg or better? And is it technically possible to nest multiple VR domains within the larger global domain, for example to explore the impact of better resolution on precip and SMB over regions of interest with steep coastlines (Antarctic Peninsula, Amundsen Sea Embayment etc)? Can you speculate about the degree of improvement in the simulation relative to the resolution of the nest?

The abstract is quite long. While not necessarily an issue, the authors *may* wish to consider making it more concise.

L14-15 I would argue that some of the coupling is captured. Suggest you qualify the statement with a word like "fully"

L126 at first glance and to the casual reader it could look from equations 2 and 3 that rainfall enters and leaves the system with minimal effect (also, is there a reason you use 'rain' and then 'rainfall'?). It may be worth briefly noting somewhere that rain can percolate into the snowpack and refreeze/be retained.

L148 wind speed differences from what? AWSs were not different from each other or the AWSs were not different from ERA5 / ANTSI? Clarification would be helpful here.

Section 3.1 – the heading is the same as for section 2.2. I imagine this should be something like 'surface mass balance'

L165-167. How does ANTSI SMB compare to the models evaluated in Mottram et al? It would be good to see where it fits into that range too.

L172-173 Suggest you expand on this last point for clarity, e.g. split into two sentences: "...but not for the Antarctic Peninsula (xxx). // The higher SMB over the Antarctic Peninsula is a result of the enhanced SMB over the grounded ice sheet (xxx)."

Table 1 Do you define these acronyms (E. Ant etc.) anywhere? Apologies if I've missed them, but worth defining in the caption if not.

Fig 2 Pedantic, but you've used the notation 'GT yr-1' in the text, and 'GT / yr' on the axis labels. Suggest using just one.

L174 – 180 It kind of looks from Fig 2 like the AMIP/ANTSI SMB estimates converge over time for the East Antarctic grounded ice sheet and diverge over time for ice shelves. The latter is consistent with the negative ANTSI trend vs none for AMIP, as you note at L179. Any ideas why that might be?

L187 "discussed in results" is a bit vague… could you be more specific?

Second half of Fig 3 caption, from L644-646. I get what you're saying here – that panels c and d compare the performance of ANTSI to the Medley reconstruction and RACMO, taking AMIP as the baseline. But I struggled a bit with the wording. Is there a way you can rephrase to make it a little clearer?

e.g. something like "Panels (c) and (d) show ANTSI SMB bias relative to AMIP compared with the Medley Reconstruction and RACMO2.3, respectively, with green and purple indicating reduced/increased bias in ANSTI relative to the dataset compared against?"

L195-198 I'm finding this sentence hard to get my head around. I wonder if it would be clearer to talk about higher and lower biases rather than 'agreement', as I struggle to figure out from this sentence how the three datasets (AMIP, ANTSI, RACMO) relate to one another. Unless I've misunderstood and you're talking about how much RACMO and ANTSI dis/agree with each other??

Figure 5 caption – it would be helpful to simply state that negative values/blue colours mean that AMIP biases are larger than ANTSI and that positive values/red colours mean that ANTSI biases are larger than AMIP compared to QSCAT (I hope I've got that the right way around!)

Figure 6 –

1) what do grey colours mean?
2) Please define the acronyms in the figure labels (e.g. "relative change (%) in large-scale precipitation ('LS precip') and integrated meridional water vapor transport ('vIVT')…")

L230 It might help the reader to briefly expand on why having a smaller time step impacts microphysics and hence precipitation totals.

Figure 7 –

1) y axes of panels a) and b) are labelled 'mmwe' – presumably this should be 'g m-2' or similar?
2) Again, would be helpful to state what the grey colours mean on panels e) and f)
3) L662-663 – it might be clearer to just say "relative difference between ANTSI and AMIP total cloud liquid water path…" (same applies to the phrasing about diffs between ANTSI/ERA5)

L263-264 "(though the latter is a product of the specific microphysical scheme)" – I think you need to clarify here, because it sounds like you're saying it's only a model thing. The impacts on cloud cover are also potentially seen in the real world, in which case I suggest something like "(though the latter is a product of the specific microphysical conditions")" – or you can limit it to model-world by qualifying the statement, e.g. "(though in the model the latter is a product of the specific microphysical scheme)"

L271 it isn't immediately clear that the 1.59 ±1.13°C refers to winter biases. Suggest rewording: "…winter biases, which average 1.59±1.13°C…"

Figure 9 caption – "comparing" is in the second sentence twice. Suggest removing the first.

Figure 10 – again worth noting what the grey colours mean (or just do it once and then say 'as in fig x…')

L332 – In a sentence or two, can you briefly summarise the key highlights of the comparison with ERA5?

L340-342 It's not clear what the positive and negative biases refer to – presumably these refer to sensible heat biases when the flux is positive and negative (?). Please note this explicitly (the same applies later, although once you've stated it once I think that should suffice).

L354 It would be helpful to clarify that you mean positive LW biases.

L355-356 I understand form what you're saying that LWP is not the only driver of LWdown. Might it be worth stating this more directly? i.e. something like "… and other factors also influence LWd" ?

L367-370 Splitting this sentence into two would help its readability. Suggest splitting after the Weddell Sea bit, eg.: "the Weddell Sea sector (nearly 15 W m-2 in some regions). The latter is consistent with cloud clearing…"

Second para pp 13 - Can you comment on the source of the increased LWP (and hence LW/SW) bias over the Southern Ocean in ANTSI? How does it compare to cloud + SEB biases in other models? As you say, the IWP bias has been reduced over this region, while as far as I understand in many models it is the over-enthusiastic production of ice at the expense of liquid by the microphysics (i.e. the vapour deposition & the Wegener-Bergeron-Findeisen process) that causes the Southern Ocean bias. How much of a role does the microphysics and/or large-scale cloud scheme have in driving this?

Second para pp 14 – I think it's worth noting / emphasising that VR ESMs like this are not going to be used for the same purposes as RCMs. The strengths of a VR ESM are to simulate the coupled response of the climate and the interactions between various elements over the long-term, e.g. for sea level rise / large-scale climate factors like precip/temp and how that relates to (S)MB. RCMs are don't have the same level of interactivity or capacity to perform extremely long simulations easily.

How much tuning is done? And does the relatively smaller improvement compared with those in Greenland reflect this?

**Supplement**

Table S1 – the surface melt values are given in mmwe but the trends are shown in GT/yr/yr.

Fig S3 – Water paths in mmwe?

Table S2 "near surface" → "near-surface"

Fig S4 "near-surface-temperature" → "near-surface temperature" (for consistency). Colourbar for the top row is very small and half of the 'd' in 'wind speed' is cut off. Rogue vorticity symbol (?) above the (n) label. As with other figures, note what the grey colours indicate. Also need to be consistent in labelling of "near-surface" for wind and temperature.

Fig S5 labels on panels (j) and (f) have been cut off. Top colourbar is again too small to read. As above, need to be consistent in labelling of "near-surface" for wind and temperature.

Fig S6 "directed at" → "directed towards"?

Fig S8 see other comments re grey areas

**Technical corrections**

L54 "Variable-resolution(VR)  grids" → "Variable-resolution (VR) grids" (space in the wrong place)

L98 Feel like this is missing a few words, what about e.g. "while a nominal 1° (standard) grid is used in the exterior"

L117 "dycore" → "core"

L137 "drifting snow scheme" ?

L172 "but not for the Antarctic Peninsula"

L179 "an positive trend" → "a positive trend"

L213 "in" → "on" some ice shelves?

L242 extra space between "within the" and "75S"

L260 extra space before ".  Cloud properties"

L295 extra space between "thus  generating"

L303 extra space between "improved , "

L324 space missing between "longwave radiation,(LWdown…)"

L327 double full stop after "coherent".

L363 extra space between 'W m-2' and 'over'

L364 "LWdown" → "LW$_{down}$" (also L367)

L368 "the East Antarctica" → "the East Antarctic" / "East Antarctica"

L375 missing an "and" after the brackets?

L376 "standard" → "standard deviation"

L411 typo in "also"

---

## Referee Comment (RC2)

Review of
*Evaluating the Impact of Enhanced Horizontal Resolution over the Antarctic Domain Using a Variable-Resolution Earth Systems Model*
by Tri Datta et al.

General comments :

This paper presents the evaluation of a variable-resolution configuration of the CESM general circulation model (ANTSI) to represent the Antarctic climate over the past decades.
An in-depth assessment of the surface mass balance and energy budget is performed together with a systematic comparison with the standard model configuration (1° regular grid).
The work is serious, the paper is overall well written and the figures are well crafted.
Although I do recognise the merit of this work and even though I overall enjoyed reading this manuscript, I have very major concerns regarding the current version of the paper and unfortunately, I cannot support its publication in the present state. However, I encourage the authors to complement their work. Please find herebelow my major remarks followed by minor comments.

- Although the authors advocate in the conclusion that 'this work suggests that the variable-resolution setup over Antarctica can be a valuable tool for representations of precipitation, surface mass balance ...', using such a simulation configuration for studying polar climates is absolutely not new. Refining locally the grid of a GCM over Antarctica was the approach and methodology of a number of studies, among which pioneering papers by Krinner et al. (Krinner et al. 1997a, Krinner et al. 1997b, Krinner et al. 2007, Krinner et al. 2014). These papers should be cited in the manuscript but besides this, I am therefore wondering what is the real scientific contribution of the present study, beyond the evaluation of a specific configuration of CESM. I'll leave it to the editors to decide if the content of the paper is sufficient to warrant publication

Krinner, G., C. Genthon, Z.-X. Li, and P. Le Van, 1997a: Studies of the Antarctic climate with a stretched-grid general circulation model. J. Geophys. Res., 102, 13 731–13 745, doi:10.1029/96JD03356.

Krinner G, Genthon C (1997b) The Antarctic surface mass balance in a stretched grid general circulation model. Ann Glaciol 25:73–78

Krinner, G., O. Magand, I. Simmonds, C. Genthon, and J.-L. Dufresne, 2007: Simulated Antarctic precipitation and surface mass balance at the end of the 20th and 21st centuries. Climate Dyn., 28, 215–230, doi:10.1007/s00382-006-0177-x.

Krinner, G., Largeron, C., Ménégoz, M., Agosta, C., & Brutel-Vuilmet, C. (2014). Oceanic forcing of Antarctic climate change: A study using a stretched-grid atmospheric general circulation model. Journal of Climate, 27(15), 5786–5800. https://doi.org/10.1175/JCLI-D-13-00367.1

- A variable-resolution GCM is a very powerful tool to study climate at the regional scale and how local features depend on and affect the global climate. However, a climatic study with such a model configuration (without nudging) is fully relevant only if the climate is satisfactorily reproduced at the global scale. This is especially true if one wants to run climate scenario simulations. I thus have the followings questions : How is the global circulation reproduced in ANTSI ? How do global temperatures and radiative fluxes at the top-of-the-atmosphere compare to CESM2-AMIP ? Was a parameters re-tuning necessary for ANTSI ?
Furthermore, as the study focuses on the Antarctic, one may question how the statistics of the climate indices (SAM...) and the main large-scale circulation patterns (wavenumber-3 pattern, ...) which are relevant for the Antarctic climate compare between the two simulations. Part of the answer is already in the supplementary materials.

- Throughout the analysis, it is extremely difficult to disentangle the effect of the enhanced resolution over the Antarctic from that of the change in dynamical core (and therefore change in large scale circulation). This is particularly critical for temperature and melting but also for precipitation which strongly depends on both large scale dynamics and fine scale topographical features. As a consequence, the comparison between ANTSI and CESM2 AMPI is often not fully conclusive or not completely convincing. I would suggest the authors to run an additional simulation using for instance the regular 1° resolution configuration with nudging towards ANTSI down to ~ 60°S (in order to simulate a similar Southern Ocean storm-track and large scale maritime advections towards the ice sheet). See Genthon et al. (2002) for an example.

Genthon, C., Krinner, G., & Cosme, E. (2002). Free and Laterally Nudged Antarctic Climate of an Atmospheric General Circulation Model, *Monthly Weather Review*, *130*(6), 1601-1616.

- Several pieces of literature suggest that ERA5 is the best reanalysis product to represent certain aspects of the Antarctic climate. However, it is definitely not a reference product for a number of variables. First and foremost, ERA5 can absolutely not be used to evaluate the cloud liquid and ice water content in Antarctic clouds (see for instance Silber et al. 2019, Vignon et al. 2021).
Regarding the surface wind, ERA5 strongly underestimates the winter wind speed in the interior and in coastal regions of the Antarctic (Gossart et al. 2019) and I would strongly recommend the authors to use another reference dataset (the AWS network for example).

Gossart, A., Helsen, S., Lenaerts, J. T. M., Broucke, S. V., van Lipzig, N. P. M., & Souverijns, N. (2019). An Evaluation of Surface Climatology in State-of-the-Art Reanalyses over the Antarctic Ice Sheet, *Journal of Climate*, *32*(20), 6899-6915.

Silber, I., Verlinde, J., Wang, S., Bromwich, D. H., Fridlind, A. M., Cadeddu, M., Eloranta, E. W., & Flynn, C. J. (2019). Cloud Influence on ERA5 and AMPS Surface Downwelling Longwave Radiation Biases in West Antarctica, *Journal of Climate*, *32*(22), 7935-7949.

Vignon, É., Alexander, S. P., DeMott, P. J., Sotiropoulou, G., Gerber, F., Hill, T. C. J., et al. (2021). Challenging and improving the simulation of mid-level mixed-phase clouds over the high-latitude Southern Ocean. *Journal of Geophysical Research: Atmospheres*, 126, e2020JD033490. https://doi.org/10.1029/2020JD033490

**Minor comments :**

l24 : Acronym VR-CESM2 not introduced yet.
L50 : Coupling with ice-sheet models is not very common.
l56 : Note that two-way nesting is possible with certain RCMs.
Section 2.1 : Can you expand a bit more on the physical content of the model (for the relevant parametrisations). In particular, can you give more details on the surface snow scheme ?
Section 2.1.1. 32 vertical levels is a coarse resolution. What is the model top height ? What is the resolution near the surface in the boundary-layer ? In the mid troposphere ? Is it sufficient to capture the katabatic flow correctly ? Same question for boundary-layer clouds ?
L107 : Storage ? Do you mean cpu time ?
L113 : In line with one of my major comment : is the tuning of ANTSI similar to that of CESM2 in the standard configuration ?
L187 : 'discussed in Results' : we are already in the Results section.
L294-297 : Please cite the external literature here.

---

## Author Response (AR1)

Dear Editors,

We have revised a version of the manuscript in accordance with the response to the reviewers (specific responses are included withing this document. In addition to the minor changes, we have added an author as well as changes in the Discussion/and Introduction/Methods added primarily for clarity. Mainly, we have added an additional section (3.3.3) and 2 supplemental figures (Supplemental Figures 8,9) and a table (Supplemental Table 5) to address Reviewer 2's main concern about whether (a) the differences were a product of resolution or dynamical core (b) the replication of large-scale circulation patterns. We believe that these additions will add to the completeness of this work and address the main concerns there.

Thank you for your patience.

Sincerely,

Rajashree Tri Datta
* * *
From Review #1

Have you tested different VR nests? e.g. 0.1 deg or better? And is it technically possible to nest multiple VR domains within the larger global domain, for example to explore the impact of better resolution on precip and SMB over regions of interest with steep coastlines (Antarctic Peninsula, Amundsen Sea Embayment etc)? Can you speculate about the degree of improvement in the simulation relative to the resolution of the nest?
We haven't tested resolutions that high -this may be possible in the future, but as you well know, the difficulty at very high resolution with intense topography is the hydrostatic assumption. In the future, the possibility of nesting a non-hydrostatic model inside a hydrostatic model may advance this nesting more fruitfully.

The abstract is quite long. While not necessarily an issue, the authors may wish to consider making it more concise.
Unfortunately, in response to additional critiques, the abstract is now actually longer. My apologies.

L14-15 I would argue that some of the coupling is captured. Suggest you qualify the statement with a word like "fully"
Fixed

L126 at first glance and to the casual reader it could look from equations 2 and 3 that rainfall enters and leaves the system with minimal effect (also, is there a reason you use 'rain' and then 'rainfall'?). It may be worth briefly noting somewhere that rain can percolate into the snowpack and refreeze/be retained.

Adjusted to state "Rainfall" in both and have added a sentence (L 131) to clarify the treatment of refreezing and retention

L148 wind speed differences from what? AWSs were not different from each other or the AWSs were not different from ERA5 / ANTSI? Clarification would be helpful here.

Windspeed has been added back into the comparisons, thus this sentence has changed.

Section 3.1 – the heading is the same as for section 2.2. I imagine this should be something like 'surface mass balance'

Fixed.

L165-167. How does ANTSI SMB compare to the models evaluated in Mottram et al? It would be good to see where it fits into that range too.

Good call!! Added on L175

L172-173 Suggest you expand on this last point for clarity, e.g. split into two sentences: "...but not for the Antarctic Peninsula (xxx). // The higher SMB over the Antarctic Peninsula is a result of the enhanced SMB over the grounded ice sheet (xxx)."

Arranged slightly differently for the same effect.

Table 1 Do you define these acronyms (E. Ant etc.) anywhere? Apologies if I've missed them, but worth defining in the caption if not.

I did not, but used the full names instead to avoid confusion.

Fig 2 Pedantic, but you've used the notation 'GT yr-1' in the text, and 'GT / yr' on the axis labels. Suggest using just one.

Fixed

L174 – 180 It kind of looks from Fig 2 like the AMIP/ANTSI SMB estimates converge over time for the East Antarctic grounded ice sheet and diverge over time for ice shelves. The latter is consistent with the negative ANTSI trend vs none for AMIP, as you note at L179. Any ideas why that might be?

I think the main reason from this has to do with the higher sensitivity in ANTSI to directing moisture towards the continent (and especially towards the grounded ice sheet, discussed in the Results. The aggregate effect is that there' a divergence in behavior depending on latitude.

L187 "discussed in results" is a bit vague… could you be more specific?

This is now pointing to section 3.2 (Precipitation) where this I discussed more rigorously

Second half of Fig 3 caption, from L644-646. I get what you're saying here – that panels c and d compare the performance of ANTSI to the Medley reconstruction and RACMO, taking AMIP as the baseline. But I struggled a bit with the wording. Is there a way you can rephrase to make it a little clearer?
e.g. something like "Panels (c) and (d) show ANTSI SMB bias relative to AMIP compared with the Medley Reconstruction and RACMO2.3, respectively, with green and purple indicating reduced/increased bias in ANSTI relative to the dataset compared against?"

I like that a lot better. Fixed.

L195-198 I'm finding this sentence hard to get my head around. I wonder if it would be clearer to talk about higher and lower biases rather than 'agreement', as I struggle to figure out from this sentence how the three datasets (AMIP, ANTSI, RACMO) relate to one another. Unless I've misunderstood and you're talking about how much RACMO and ANTSI dis/agree with each other??

Agreed. Stating "lower biases" is more accurate; I think it's a lot clearer this way.

Figure 5 caption – it would be helpful to simply state that negative values/blue colours mean that AMIP biases are larger than ANTSI and that positive values/red colours mean that ANTSI biases are larger than AMIP compared to QSCAT (I hope I've got that the right way around!)

Agreed. Clarified by simply stating "red values indicating higher biases"

Figure 6 –
1) what do grey colours mean?
2) Please define the acronyms in the figure labels (e.g. "relative change (%) in large-scale precipitation ('LS precip') and integrated meridional water vapor transport ('vIVT')…")

Added text for both to clarify

L230 It might help the reader to briefly expand on why having a smaller time step impacts microphysics and hence precipitation totals.

After consultation with co-authors, we've changed this to reference resolution only. The time step is actually chosen to match up with the highest resolution used – in this case, 0.25° (this can be relaxed, but may generate errors).

Figure 7 –
1) y axes of panels a) and b) are labelled 'mmwe' – presumably this should be 'g m-2' or similar?

- actually this is accurate as mmwe == kg m-2

2) Again, would be helpful to state what the grey colours mean on panels e) and f)

Added

3) L662-663 – it might be clearer to just say "relative difference between ANTSI and AMIP total cloud liquid water path…" (same applies to the phrasing about diffs between ANTSI/ERA5)

Changed

L263-264 "(though the latter is a product of the specific microphysical scheme)" – I think you need to clarify here, because it sounds like you're saying it's only a model thing. The impacts on cloud cover are also potentially seen in the real world, in which case I suggest something like "(though the latter is a product of the specific microphysical conditions)" – or you can limit it to model-world by qualifying the statement, e.g. "(though in the model the latter is a product of the specific microphysical scheme)"

Accepted the second suggestion

L271 it isn't immediately clear that the 1.59 ±1.13°C refers to winter biases. Suggest rewording: "…winter biases, which average 1.59±1.13°C…"

Agreed. Changed.

Figure 9 caption – "comparing" is in the second sentence twice. Suggest removing the first.

Changed

Figure 10 – again worth noting what the grey colours mean (or just do it once and then say 'as in fig x…')

Added same phrase in each figure just for clarity "Grey indicates where biases are not significant as compared to AMIP ensemble std dev."

L332 – In a sentence or two, can you briefly summarise the key highlights of the comparison with ERA5?

A brief 2 sentences have been added within the introductory section of 3.4 to summarize the comparison.

L340-342 It's not clear what the positive and negative biases refer to – presumably these refer to sensible heat biases when the flux is positive and negative (?). Please note this explicitly (the same applies later, although once you've stated it once I think that should suffice).

We've made this explicit by stating the direction of sensible heat flux (towards the surface)

L354 It would be helpful to clarify that you mean positive LW biases.

Added

L355-356 I understand form what you're saying that LWP is not the only driver of LWdown. Might it be worth stating this more directly? i.e. something like "… and other factors also influence LWd" ?

Altered to state "Added "downward longwave radiation is a product of many factors other than cloud liquid water path"

L367-370 Splitting this sentence into two would help its readability. Suggest splitting after the Weddell Sea bit, eg.: "the Weddell Sea sector (nearly 15 W m-2 in some regions). The latter is consistent with cloud clearing…"
Agreed. Altered.

Second para pp 13 - Can you comment on the source of the increased LWP (and hence LW/SW) bias over the Southern Ocean in ANTSI? How does it compare to cloud + SEB biases in other models? As you say, the IWP bias has been reduced over this region, while as far as I understand in many models it is the over-enthusiastic production of ice at the expense of liquid by the microphysics (i.e. the vapour deposition & the Wegener-Bergeron-Findeisen process) that causes the Southern Ocean bias. How much of a role does the microphysics and/or large-scale cloud scheme have in driving this?
This will be touched on in additional analysis which will be added to the manuscript about the impact of dycore vs resolution. This is essentially a product of enhanced vertical motion produced by the enhanced resolution, thus producing more moisture in the atmosphere overall. The overall impact is that the ice vs liquid drying happens over summer (which is already discussed, and discussed further in Herrington et al., 2022), but both ice & liquid water path are affected in winter. I think this has been discussed sufficiently in Herrington et al., 2022.

Second para pp 14 – I think it's worth noting / emphasising that VR ESMs like this are not going to be used for the same purposes as RCMs. The strengths of a VR ESM are to simulate the coupled response of the climate and the interactions between various elements over the long-term, e.g. for sea level rise / large-scale climate factors like precip/temp and how that relates to (S)MB. RCMs are don't have the same level of interactivity or capacity to perform extremely long simulations easily.
Added the phrase "thus RCMs and VR-ESMs currently serve different purposes".

How much tuning is done? And does the relatively smaller improvement compared with those in Greenland reflect this?
Actually, no tuning was done here at all. This analysis may influence future tuning in CESM3

Supplement

Table S1 – the surface melt values are given in mmwe but the trends are shown in GT/yr/yr.
Fixed

Fig S3 – Water paths in mmwe?
Yes., this is intentional. Mmwe is equivalent to $kg/m^2$

Table S2 "near surface" → "near-surface"

Fixed

Fig S4 "near-surface-temperature" → "near-surface temperature" (for consistency). Colourbar for the top row is very small and half of the 'd' in 'wind speed' is cut off. Rogue vorticity symbol (?) above the (n) label. As with other figures, note what the grey colours indicate. Also need to be consistent in labelling of "near-surface" for wind and temperature.

Added note on grey colors for all figures in supplemental

Fig S5 labels on panels (j) and (f) have been cut off. Top colourbar is again too small to read. As above, need to be consistent in labelling of "near-surface" for wind and temperature.

Fixed

Fig S6 "directed at" → "directed towards"?

Fixed

Fig S8 see other comments re grey areas

Fixed

L54 "Variable-resolution(VR) grids" → "Variable-resolution (VR) grids" (space in the wrong place)

Fixed

L98 Feel like this is missing a few words, what about e.g. "while a nominal 1° (standard) grid is used in the exterior"

Fixed

L117 "dycore" → "core"

Fixed

L137 "drifting snow scheme" ?

Fixed

L172 "but not for the Antarctic Peninsula"

Separated sentence for emphasis

L179 "an positive trend" → "a positive trend"

Fixed

L213 "in" → "on" some ice shelves?

Fixed

L242 extra space between "within the" and "75S"

Fixed

L260 extra space before ". Cloud properties"

Fixed

L295 extra space between "thus generating"

Fixed

L303 extra space between "improved , "

Fixed

L324 space missing between "longwave radiation,(LWdown…)"

Fixed

L327 double full stop after "coherent".

Fixed

L363 extra space between 'W m-2' and 'over'

Fixed

L364 "LWdown" → "LWdown" (also L367)

Fixed

L368 "the East Antarctica" → "the East Antarctic" / "East Antarctica"

Fixed

L375 missing an "and" after the brackets?

Fixed

L376 "standard" → "standard deviation"

Fixed

L411 typo in "also"

Fixed

From Reviewer #2

**1) Previous work with a stretched grid vs. refined resolution:**

Our interpretation of the main critique here is that, as a stretched grid format has been previously presented by earlier work, a variable-resolution grid over Antarctica is not new. While a revised version of the paper will discuss the difference in more detail (with better referencing and placing in the context of previous work), the key differences are noted here:

a)        This is not a stretched grid (or a nested grid), but a regional refinement – adding resolution over the most salient region of study (rather than redistributing resolution), though like the stretched grid, this preserves two way interactions over the globe. This means that it can more easily be linked to studies with the standard 1° resolution used in, say, CESM2 (although this is with the standard FV dycore).

b)        In this case, the enhanced resolution is applied over both Antarctic continent and the Southern Ocean domain, thus capturing regions with evolving sea ice conditions (a regional refinement which will be valuable for future work)

c)        Finally, the grid refinement here approaches the 0.25° resolution of reanalysis products as well as higher-resolution RCMs (e.g. RACMO or MAR). This is particularly relevant to regions with variations in fine-scale topography (e.g. the Antarctic Peninsula and portions of W. Antarctica). By comparison, the previous work with a stretched-grid approached a maximum resolution of 1° below 60° and a minimum resolution as high as 5° (see Fig. 1 from Genthon et al., 2002) or a maximum resolution of 60km (Krinner et al., 2014). While enhancing resolution itself in an ESM is not new, this manuscript introduces a regional refinement over Antarctica approaching that of RCMs.

**2) References to earlier work using a stretched grid:**

This was an oversight on our part which will be corrected in the revised version the paper.

**3) A more rigorous examination of general circulation**

We agree that the manuscript would benefit from a more in-depth analysis of key features of general circulation. We have begun some additional diagnostics from what was already in the supplemental material.

**4) A disambiguation of the effects of dycore vs resolution:**

We omitted this discussion from this paper primarily because the relative impact of a change in dycore vs resolution in a polar region is discussed extensively in Herrington et al., 2022, although this in Greenland. However, similar experiments have been done over Antarctica and these will be analyzed and discussed in a revised version.

**5) The use of AWS stations to discuss near-surface winds**

A comparison between the ANTSI grid and AWS data was in a draft version of the manuscript (as in Fig.9b, but for wind) and will be included in the revised manuscript for thoroughness.

References:

Genthon, C., Krinner, G., & Cosme, E. (2002). Free and Laterally Nudged Antarctic Climate of an Atmospheric General Circulation Model. *Monthly Weather Review*, *130*(6), 1601–1616. https://doi.org/10.1175/1520-0493(2002)130<1601:FALNAC>2.0.CO;2

Herrington, A. R., Lauritzen, P. H., Lofverstrom, M., Lipscomb, W. H., Gettelman, A., & Taylor, M. A. (2022). Impact of grids and dynamical cores in CESM2.2 on the surface mass balance of the Greenland Ice Sheet. *Journal of Advances in Modeling Earth Systems*, *n/a*(n/a), e2022MS003192. https://doi.org/10.1029/2022MS003192

Krinner, G., Largeron, C., Ménégoz, M., Agosta, C., & Brutel-Vuilmet, C. (2014). Oceanic Forcing of Antarctic Climate Change: A Study Using a Stretched-Grid Atmospheric General Circulation Model. *Journal of Climate*, *27*(15), 5786–5800. https://doi.org/10.1175/JCLI-D-13-00367.1

l24 : Acronym VR-CESM2 not introduced yet.
Fixed.

L50 : Coupling with ice-sheet models is not very common.
l56 : Note that two-way nesting is possible with certain RCMs.
Section 2.1 : Can you expand a bit more on the physical content of the model (for the relevant parametrisations). In particular, can you give more details on the surface snow scheme ?
This is addressed in referenced literature

Section 2.1.1. 32 vertical levels is a coarse resolution. What is the model top height ? What is the resolution near the surface in the boundary-layer ? In the mid troposphere ? Is it sufficient to capture the katabatic flow correctly ? Same question for boundary-layer clouds ?
This is beyond the scope of this study

L107 : Storage ? Do you mean cpu time ?
No, this refers to storage of the actual outputs

L113 : In line with one of my major comment : is the tuning of ANTSI similar to that of CESM2 in the standard configuration ?
Yes, it is. As listed, no tuning was performed

L187 : 'discussed in Results' : we are already in the Results section.
Excellent point. This has been addressed.

L294-297 : Please cite the external literature here.

This is added

---

## Author Response (AR2)

Dear TC,

Thank you very much for the response.
I agree that the use of "RACMO" was ambiguous and I have replaced all instances with RACMO with RACMO.3p2 as requested (it seemed the most detailed). I've also altered the coloring in the Supplemental, as requested and checked through figures.

Thank you for your time and attention.

Sincerely,

Rajashree Tri Datta